# Synergistic H&E and IHC image analysis by AI predicts cancer biomarkers and survival outcomes in colorectal and breast cancer
Yating Cheng[1,2], Norsang Lama[1,2], Ming Chen[1] ✉, Eghbal Amidi[1], Mohammadreza Ramzanpour[1],
Md Ashequr Rahman[1], Joanne Xiu[1], Anthony Helmstetter [1], Lauren Dickman[1], Jennifer R. Ribeiro [1],
Hassan Ghani [1], Matthew Oberley [1], David Spetzler [1] & George W. Sledge [1] ✉

## Abstract

**Background** Recent advancements in immunotherapy, particularly pembrolizumab, have shown promising results in treating metastatic colorectal cancer (CRC) and triple-negative breast cancer (TNBC). Accurate detection of predictive biomarkers, such as microsatellite instability (MSI)/mismatch repair deficiency (MMRd) and programmed death-ligand 1 (PD-L1), is key to efficacy of these treatments. Traditional methods like immunohistochemistry (IHC) and next-generation sequencing are effective but are labor intensive and require subjective interpretation.

**Methods** We developed a dual-modality transformer-based model for predicting MSI/MMRd and PD-L1 status using hematoxylin & eosin and IHC stained whole slide images. We evaluated the model using area under the receiver operating curve (AUROC). Time-on-treatment (TOT) and overall survival (OS) were derived from insurance claims and analyzed by Kaplan–Meier method. Hazard ratios (HR) were determined using the Cox proportional hazard model.

**Results** Our AI framework achieves clinical-grade performance, with AUROC exceeding 0.97 for MSI/MMRd prediction in CRC and 0.96 for PD-L1 prediction in breast cancer. Patients with biomarker-positive model predictions demonstrated prolonged TOT and OS when treated with pembrolizumab. For breast cancer patients, the model's predictions were superior to PD-L1 IHC in stratifying patients with improved outcomes on pembrolizumab, suggesting a reevaluation of existing PD-L1 status thresholds.

**Conclusions** This study promotes the integration of advanced AI tools in clinical pathology, aiming to enhance the precision and efficiency of cancer biomarker evaluation and offering a customizable framework for varied clinical scenarios. Our model enhances predictive accuracy, integrating features from both staining methods, and exhibits superior prognostic precision compared to current biomarker assessments.

## Plain language summary

Current methods to identify cancer patients who will respond to specific immunotherapy treatments are labor intensive and require interpretation of markers in cancer tissue by clinicians. We developed a computer model that analyzes tumor tissue images to predict the status of key biomarkers that are used to select patients with colorectal cancer and triple-negative breast cancer for pembrolizumab treatment. We show that our model predicts statuses with high accuracy and identifies patients with improved outcomes on pembrolizumab. Clinical adoption of this tool could improve the precision and efficiency of cancer patient evaluation and aid clinical decision making.

Recent advancements in immunotherapy, notably the application of pembrolizumab—an anti-PD-1 antibody—have demonstrated remarkable efficacy in treating various cancer types, including microsatellite instability-high (MSI-H)/mismatch repair deficient (MMRd) metastatic colorectal cancer (CRC) and programmed death-ligand 1 (PD-L1)-positive triple-negative breast cancer (TNBC), significantly prolonging progression-free and overall survival[1,2].

The conventional detection of MSI often relies on polymerase chain reaction (PCR)-based methods[3], although next-generation sequencing (NGS)[4] is increasingly utilized for comprehensive profiling, while MMRd and PD-L1 status are routinely assessed by immunohistochemistry (IHC)[5]. IHC, while extensive in its application, remains labor-intensive, requiring substantial time investment from highly trained pathologists

[1]Caris Life Sciences, Phoenix, AZ, USA. [2]These authors contributed equally: Yating Cheng, Norsang Lama.
✉e-mail: mchen@carisls.com; gsledge@carisls.com

**Table 1 | Patient characteristics of the study cohorts analyzed**

| | CRC (MMRd), N (%) | CRC (MSI), N (%) | BRCA (PD-L1), N (%) |
|---|---|---|---|
| Age | | | |
| ≥65 years | 9523 (46%) | 9572 (46%) | 6643 (44%) |
| <65 years | 11,297 (54%) | 11,307 (54%) | 8530 (56%) |
| Sex | | | |
| Male | 11,468 (55%) | 11,512 (55%) | 202 (1%) |
| Female | 9352 (45%) | 9367 (45%) | 14,971 (99%) |
| Biomarker status | Deficient: 1542 (7%) | MSI: 1662 (8%) | Positive: 2631 (17%) |
| | Proficient: 19,278 (93%) | MSS: 19,217 (92%) | Negative: 12,542 (83%) |

*BRCA* breast cancer, *CRC* colorectal cancer, *MMRd* mismatch repair deficiency, *MSI* microsatellite instability, *MSS* microsatellite stable.

for interpretation. The process is not only time-consuming but also subject to inter-observer variability, which can affect diagnostic consistency and accuracy[6]. Similarly, NGS offers a high-resolution view of the genetic landscape of tumors but also demands complicated workflows and expert interpretation[7].

With advances in computational algorithms and tools, considerable investment has been made in the development of artificial intelligence (AI) tools to aid in the clinical assessment of biomarkers and outcomes[8-21]. Given that hematoxylin & eosin (H&E) staining is the most widely used and cost-effective method in clinical settings, researchers have focused extensively on developing tools for analyzing H&E-stained whole slide images (WSIs) to predict biomarker status[22-26]. For example, deep learning-based predictive algorithms have been successful in extrapolating PD-L1 status from H&E-stained images in breast cancer (BRCA)[27]. Wagner et al. introduced an end-to-end transformer-based model for CRC biomarker prediction from H&E WSI in 2023[22]. On the other hand, limited studies have been conducted on AI-assisted IHC interpretations[28-31]. For instance, a weakly supervised deep learning model on raw IHC images was utilized to predict PD-L1 status in non-small cell lung cancer (NSCLC). This model achieved AUROC scores of 0.80–0.88 and led to an improved association with response to immune checkpoint inhibition[29]. In another study, Huang et al. developed a WSI features extraction pipeline using both H&E and multiplex IHC images to predict neoadjuvant chemotherapy (NAC) outcomes in patients with HER2-positive (HER2+) BRCA and TNBC[32].

Despite these advances, few studies have systematically evaluated whether combining H&E and IHC images through a unified AI framework improves biomarker prediction or enhances prognostic relevance. In our study, we developed DuoHistoNet, an efficient dual-modality transformer[22]-based framework that integrates both H&E and IHC stained images for enhanced biomarker prediction. We hypothesized that (1) dual-modality models integrating H&E and IHC images would achieve superior performance in biomarker prediction compared to single-modality models, and (2) AI-predicted biomarker status would demonstrate prognostic value comparable to—or exceeding—that of pathologist-annotated status. Our results demonstrate that the dual-modality approach enables precise predictions of biomarkers, specifically MSI/MMRd status and PD-L1 status, exhibiting prognostic stratification that is comparable to, or even surpasses, that of the actual biomarker statuses detected through conventional methods. Our system provides a flexible and comprehensive framework that can adapt to the available data types, be it H&E, IHC, or both. It also features the capability to customize predictive thresholds to fit a range of clinical scenarios. Our model has strong potential for clinical application, promising to aid decision-making in oncology and to contribute to more personalized patient care. With its robust performance and adaptability, this model could be an invaluable resource in clinical

settings, enhancing patient outcomes through precise, biomarker-driven therapies.

## Methods

### Patient samples
For this study, we used a consecutive cohort of molecularly-profiled patient samples from the Caris Life Sciences (Phoenix, AZ, USA) database between the years of 2008–2023. Inclusion criteria included cases with available WSIs and a test result for MSI/MMRd or PD-L1. Patients with <21 days of time-on-treatment (TOT) were excluded from survival analyses. Formalin-fixed, paraffin-embedded (FFPE) tissue sections were used for all assays. Specimen preservation and storage followed Caris Life Sciences' standard laboratory protocols, in accordance with their SOP for specimen retention. The detailed patient characteristics are summarized in Table 1 and Supplementary Tables 1 and 2. This study was conducted in accordance with guidelines of the Declaration of Helsinki, Belmont Report, and US Common Rule. In keeping with 45 CFR 46.104(d)(4), this study was carried out utilizing retrospective, deidentified clinical data. Therefore, this study is considered exempt from institutional review board (IRB) approval and no patient consent was necessary. Exempt status was determined by WIRB-Copernicus Group (WCG®) IRB.

### Microsatellite instability (MSI) assay and interpretation
MSI status was assessed by directly analyzing 2810 known homopolymer to pentapolymer microsatellite regions within the targeted whole exome sequencing (WES) gene panel. These regions were compared against the reference genome (hg38) available from the UCSC Genome Browser database. Microsatellite loci alterations were identified through somatic insertions or deletions, counting only those changes that affected the number of tandem repeats. Detection of genomic variants at these loci utilized the same depth and frequency criteria as our mutation detection protocols. An MSI-H status was assigned to samples with 116 or more altered loci. Samples with 113 to 115 altered loci were considered equivocal, and those with 112 or fewer were classified as microsatellite stable (MSS). Patients with an indeterminate MSI result (sample depth <500x) were not included in the study cohort.

### Mismatch repair (MMR) assay and interpretation
IHC analysis was conducted on FFPE tissue sections on glass slides per standard clinical workflows. Automated staining techniques, following the manufacturer's instructions, were applied, and validated according to CLIA/CAP and ISO standards. Board-certified pathologists independently reviewed all IHC results. MMR protein expression was determined using specific antibody clones for MLH1 (M1 antibody), MSH2 (G2191129 antibody), MSH6 (44 antibody), and PMS2 (EPR3947 antibody; Ventana Medical Systems, Inc., Tucson, AZ, USA). MMRd was indicated by the complete absence of expression for any tested protein, whereas proficient MMR (MMRp) demonstrated positive staining across all four proteins. Internal controls were utilized where possible, and every IHC slide utilized external positive and negative run controls.

### PD-L1 assay and interpretation
The PD-L1 immunohistochemical staining was carried out on FFPE sections using the PD-L1 22C3 pharmDx kit, according to FDA standards (Agilent Technologies, Santa Clara, CA, USA). This process was subjected to the same rigorous automated staining techniques, optimization, and validation protocols as established for the MMR IHC assays, in compliance with CLIA/CAP and ISO guidelines. The Combined Positive Score (CPS) was calculated with the formula: (number of PD-L1 positive cells (tumor cells, lymphocytes, and macrophages) / total number of viable tumor cells) * 100. Board-certified pathologists were trained according to the PD-L1 IHC 22c3 pharmDx Interpretation Manual and evaluated the results independently at a single institution (Caris Life Sciences). PD-L1 subgroups were defined as negative (CPS < 10) or positive (CPS ≥ 10).

## Transformer-based model training details

**Image acquisition**. H&E and IHC WSIs from the same patients were acquired for a cohort study of BRCA and CRC, including samples from both primary and metastatic sites (20,820 cases for CRC MMR cohort, 20,879 cases for CRC MSI cohort, 15,173 cases for BRCA PD-L1 cohort). WSIs were scanned using either Philips or Leica scanners at 40X resolution.

## Model setup

Our biomarker prediction framework operated through a three-stage pipeline: (1) data preprocessing, (2) feature extraction using a transformer-based model, and (3) aggregation of features to produce the final WSI-level prediction.

Preprocessing: initially, two QuPath[33] pixel classification models were trained to segment tissues from H&E and IHC WSIs respectively. An object detection model based on YOLO framework[34] was trained to detect control tissue (tissue microarray cores) on IHC WSIs and exclude it from the WSI tissue mask. Subsequently, WSIs were tessellated into $224 \times 224$-pixel tiles at $\times 10$ magnification, with a detailed resolution of ~1.0 micron per pixel, post-application of the tissue masks generated from the QuPath pixel classifiers. For Philips images in iSyntax format, a conversion to TIFF format was performed prior to processing.

Feature extraction: features for each tile were extracted utilizing the CTransPath model[35], a hybrid architecture that combines the Swin Transformer[36] with a convolutional neural network (CNN) structure. The CTransPath is equipped with three initial convolution layers that enhance local feature detection and improve training stability, succeeded by four stages of Swin Transformer layers. These transformer layers add global contextual information via self-attention modules. The CTransPath was pretrained in a self-supervised manner on a large dataset of unlabeled histopathological images from The Cancer Genome Atlas (TCGA) and Pathology AI Platform (PAIP)[37]. This feature extractor generated a 768-dimensional feature vector per tile, which was then utilized in further downstream analysis.

Aggregation and prediction: the aggregation stage ingests patch embeddings from a WSI, employing a transformer-based aggregation module with multi-headed self-attention to process the sequence of embeddings. It permits each patch to interact with every other, thereby enabling a comprehensive assessment. Our transformer-based aggregator uses a similar transformer backbone to that described by Wagner et al.[22], who demonstrated that the transformer-based aggregation has superior performance over other attention-based MIL approaches[38,39] on biomarker prediction. It mainly consists of a linear projection layer, two transformer layers, and multi-layer perceptron (MLP) head. In our DuoHistoNet framework, we used two parameter-shared transformer branches for dual H&E and IHC inputs as shown in Fig. 1. The output features from two branches were then concatenated along the channel dimension and passed to the classifier, i.e., MLP head, for final prediction. Parameter-sharing helps to reduce the number of model parameters, which subsequently results in a memory-efficient model. Furthermore, we incorporated a branch-dropout mechanism that omits one of the branches during model training, effectively enhancing the model's generalization ability.

## Experimental setup and implementation details

For all experiments, we employed a five-fold cross-validation scheme, subdividing our dataset into training, validation, and test sets in a 6:2:2 ratio, respectively. During training, the validation set was used to determine the best model, which was finally evaluated on the test set. The transformer models were trained with the AdamW optimizer using weight decay of $5* 10^{-4}$ and learning rate of $1* 10^{-5}$. All models were trained for 10 epochs with a batch size of one and a branch dropout probability of 0.3. The models were evaluated every 500 iterations for all cohorts. The number of tiles per WSI varies widely, ranging from 2 to 11,000 tiles. Approximately 80% of WSIs contain fewer than 3200 tiles, and 50% have fewer than 1000 tiles. Given the wide variation in tile

counts, we found no performance degradation when limiting the model to training with only 500 tiles. Consequently, we randomly selected a maximum of 500 tiles per epoch during training and validation phases, which reduced the GPU memory usage and allowed the model to be trained on a GPU with a limited memory capacity (16 GB). For the testing phase, all available tiles from each WSI were used and the processing was performed on a CPU. Using multiple GPUs, with one assigned to each fold, considerably sped up the training process.

## Visualization and explainability

Visualization and explainability of transformer-based models are crucial for clinicians to comprehend the decision-making process of deep learning models. To understand how our proposed DuoHistoNet analyzes WSIs for decision-making, we adopted a visualization technique from the study by Wagner et al.[22]. This method is well-suited to DuoHistoNet, as it utilizes a similar transformer-based aggregation framework for its prediction. First, to assess the influence of each individual patch on the classification score, we employed attention rollout. This technique involves the recursive multiplication of attention maps from preceding layers, offering a cumulative insight into patch impact. Additionally, we visualized the attention scores assigned by each transformer head through class token self-attention. These scores were normalized and tuned to ensure a coherent visual representation within the standardized range of [0,1]. By processing each patch independently via the transformer model, we could directly quantify their respective contributions to the classification decisions, facilitating the visualization of their impact scores within the pre-defined range. The classification heatmap was calculated by using a single tile as an input to the model. Contribution heatmaps (attention x classification score) were derived as the product of individual tile classification score and attention score. Using the methodologies above, we generated heatmaps for models trained on H&E WSI only, IHC WSI only and duet H&E + IHC WSIs. The resulting visualizations serve as an intuitive guide to understanding the areas within the tissue samples that most inform the model's predictions.

## Model evaluation

Upon completion of training, we evaluated the model's performance across each fold's test set as well as on a holdout dataset. We used the area under the receiver operator curve (AUROC) as our main evaluation metric.

## Clinical outcome analysis

Two different endpoints were assessed in this study. Pembrolizumab was given in a real-world setting, with no randomization, and outcomes were analyzed retrospectively. TOT was inferred using insurance claim data, calculated as the interval between the first and last administrations of pembrolizumab. Overall survival (OS) was defined from the initiation of pembrolizumab treatment to either the date of death in the real-world evidence (RWE) dataset or the last known contact in the insurance claims database. Patients with no claim for over 100 days were presumed deceased, while those with contact within 100 days of the last claims data refresh were considered alive but censored in the analysis[40]. In addition, for the survival analysis, patients in the holdout dataset with TOT of less than 21 days were excluded from the final analysis, as these cases were considered to represent poor quality data. Kaplan–Meier survival metrics were computed, with hazard ratios (HR) derived from the Cox proportional hazard model and $p$ values ascertained via the log-rank test.

## Statistics and reproducibility

This was a retrospective analysis based on real-world clinical data and therefore lacked randomization or blinding. The study was not prospectively powered to detect specific clinical outcomes. All statistical analyses were performed using Python. For outcome-based analyses, Kaplan–Meier survival curves were compared using the log-rank test, and HRs with 95% confidence intervals (CIs) were calculated using Cox

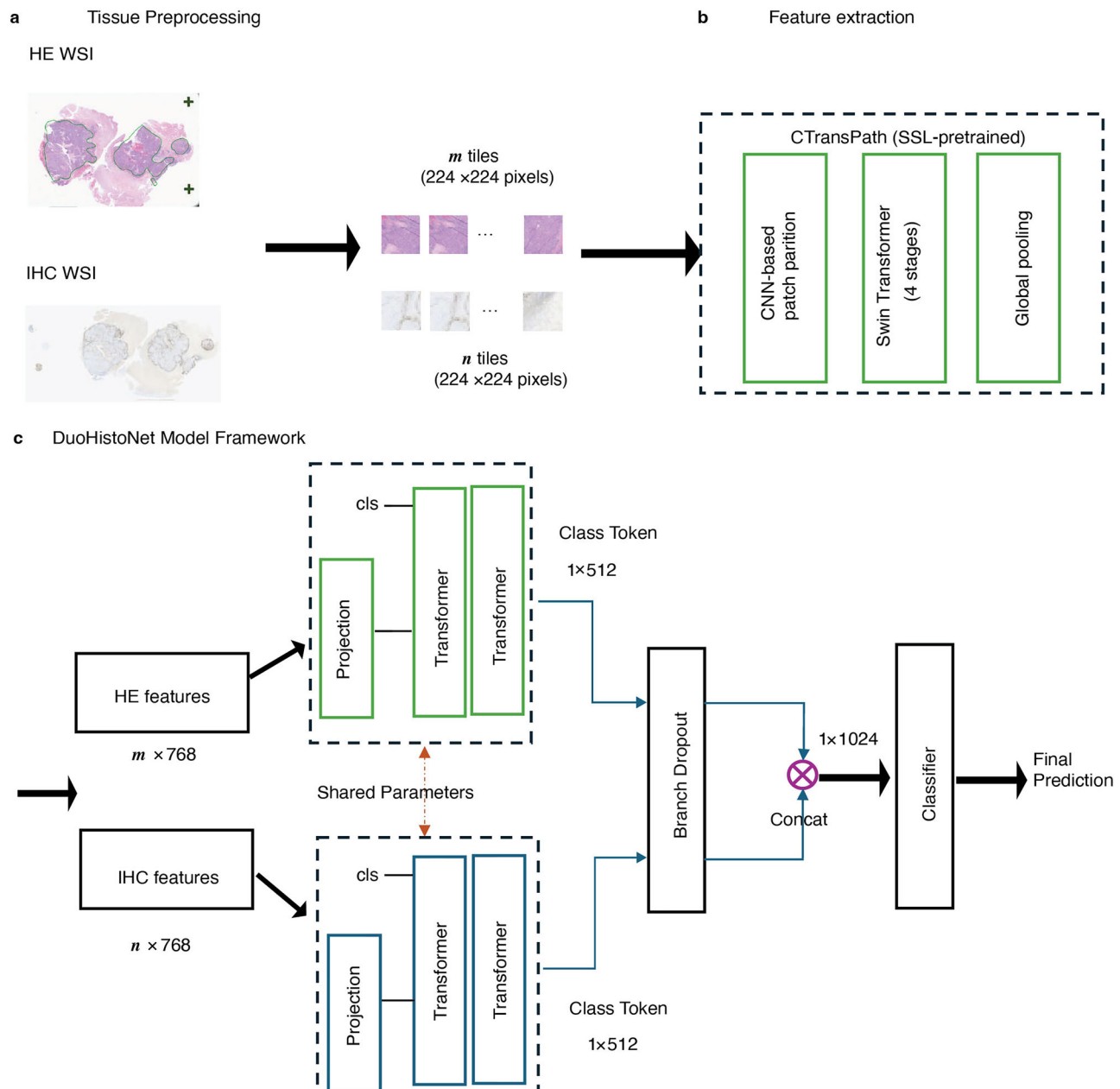

**Fig. 1 | Overview of the data pre-processing and model architecture. a** The pre-processing pipeline initiates with the digitization of whole slide images (WSIs), followed by tissue segmentation and tessellation of the WSIs into patches for analysis. **b** Illustration of the model architecture showcasing the pre-trained feature extractor, CTransPath, which processes the initial input data. **c** The transformer-based feature aggregation module where both hematoxylin & eosin (H&E) and immunohistochemistry (IHC) features are assimilated during model training, facilitating a comprehensive learning process that leverages the intricacies of both staining methods.

proportional hazards models. The primary evaluation metric for model performance was AUROC.

Sample sizes for each cohort were: 20,820 cases in the CRC MMRd cohort, 20,879 cases in the CRC MSI cohort, and 15,173 cases in the BRCA PD-L1 cohort. For prognostic survival analyses, we included pembrolizumab-treated patients from the CRC and BRCA cohorts with available biomarker status (MSI, MMRd, or PD-L1). After excluding patients with a TOT of less than 21 days, 400 CRC patients with MSI data, 401 CRC patients with MMR IHC data, and 403 BRCA patients with PD-L1 data were included in the survival analysis.

To assess reproducibility and obtain robust model performance, a nested five-fold cross-validation strategy was implemented. For each of the five outer folds, 20% of the data served as the independent test set. The remaining 80% was used for model training and hyperparameter tuning, with 60% of the total data designated for training and 20% for validation in each iteration. Final model performance was then averaged across the five independent test sets. For each training epoch, up to 500 tiles per WSI were randomly sampled to optimize computational efficiency without compromising accuracy. During testing, all available tiles were used. Training was performed using multiple GPUs, with one GPU allocated per fold. Model architecture, hyperparameters, and training procedures were kept consistent throughout.

## Reporting summary
Further information on research design is available in the Nature Portfolio Reporting Summary linked to this article.

## Results

### Patient characteristics

In this study, leveraging the extensive Caris Molecular Database encompassing tumor profiles analyzed at the Caris Life Sciences laboratory from 2008 to 2023, we identified cohorts of patients with BRCA and CRC (20,820 cases for CRC MMRd cohort; 20,879 cases for CRC MSI cohort; 15,173 cases for BRCA PD-L1 cohort). Each case had accompanying WSIs with available test results for MSI, MMRd, or PD-L1. The detailed patient characteristics are summarized in Table 1 and Supplementary Tables 1 and 2.

### Enhancements to model framework for integrated H&E and IHC image analysis

Our model framework, DuoHistoNet, introduces important advancements in integrating input from H&E and IHC images (Fig. 1). Our methodology is derived from a well-established approach detailed in Wagner et al.'s multicentric study on transformer-based biomarker prediction in CRC histology[22]. Drawing on the architecture outlined in Wagner et al.'s research[22], we have executed the following enhancements: (1) Dual-modality input pipeline: we restructured the pipeline to enable feature integration from both H&E and IHC images, offering flexibility. Users can now employ features from either or both image types for the inference of new cases. While other studies have employed vision transformer (ViT) architectures for histopathological analysis of H&E and IHC slides[41], our approach represents one of the early attempts to integrate H&E and IHC images within a single transformer-based pipeline specifically designed for biomarker prediction. (2) Efficient tile sampling: to optimize resources, our approach incorporates random tile selection in training and validation. Instead of processing all tiles from WSIs, the model selects a random subset of maximum 500 tiles for each training epoch, a strategy that maintains model performance without utilizing exhaustive data. (3) Accelerated multi-GPU training: our 5-fold training, validation, and testing setup is now empowered by multi-GPU training configurations. Each fold is processed on a dedicated GPU, considerably reducing total training duration and facilitating a more rapid turnaround of results.

### Enhanced MMRd and MSI status prediction in CRC using the duet model with combined H&E and IHC WSIs outperforming single-stain models

Pembrolizumab has been approved for treating patients with advanced CRC harboring MSI or MMRd, markers pivotal for predicting immunotherapy outcomes. Several deep-learning–based algorithms have leveraged H&E WSIs for MSI prediction in CRC using a variety of architectures[22,23,42–46]. However, while Whangbo et al. recently demonstrated the feasibility of predicting MMRd status in endometrial cancer via a multi-resolution ensemble learning approach[47], a comprehensive predictive framework specifically targeting MMRd in CRC remains less explored.

Our dual-modality model, innovatively incorporating both H&E and IHC images, specifically focuses on MLH1 status, as MLH1 promoter hypermethylation and consecutive loss of MLH1 expression is the most common reason for MMRd[48]. This system demonstrates exceptional predictive accuracy. For MMRd prediction, the model achieved AUROC scores of 0.922, 0.947, and 0.967 for H&E alone, IHC alone, and a combination of H&E/IHC, respectively. It exhibits even higher performance in MSI status prediction (0.939, 0.952 and 0.973 for H&E alone, IHC alone, and combination of H&E and IHC, respectively). The standard H&E images-based model achieved an AUROC score of 0.9, yet integrating IHC data further refined predictive capability, boasting an AUROC of ~0.97 for both MSI and MMRd prediction (Fig. 2a, b). In addition, we generated histograms to display the distribution of prediction probabilities. Notably, our duet model demonstrated a more distinct separation of probabilities compared to the models that used H&E or IHC staining images alone (Supplementary Fig. 1). This suggests an enhanced predictive performance by our integrated approach.

A stratified analysis further demonstrated our model's robustness, accommodating variances in scanner types—specifically Philips and Leica

—and specimen site types, encompassing primary and metastatic sites (Fig. 2c). The distributions of scanner types and specimen site types are summarized in Supplementary Tables 3–6. Overall, we observed minimal impact of different scanner types on the model's performance. The model performed slightly better for WSIs scanned by Philips scanner, with the maximum difference observed in H&E WSI-based MMRd prediction (averaged AUROC scores of 0.930 and 0.905 for Philips and Leica scanner scanned images, respectively). On the other hand, the specimen site had more impact on model performance, with the maximum difference observed in H&E WSI-based MMRd prediction (averaged AUROC scores of 0.940 and 0.852 for primary and metastatic specimens, respectively). Notably, the impact of specimen type was lesser on the IHC WSI-based model (Fig. 2c).

In the context of deploying our AI tools in clinical practice, we have performed a detailed trade-off analysis (Fig. 2d and Supplementary Data 1 and 2), aiming to determine the effectiveness of our AI model as a pre-screening tool in medical settings. For instance, by focusing on MSI prediction, our goal was to evaluate the potential of our model to obviate the need for MSI NGS testing in certain patients, thereby substantially lightening the workload for pathologists and technicians. For our dual-modality approach using H&E/IHC for MSI prediction, with a false negative percentage (False negative cases/Total cases × 100) of just 0.1%, our model achieved a sensitivity exceeding 98%. Among the 20,410 patients in our cohort, 11,698 (57%) were correctly classified as MSS, demonstrating the model's ability to capture a substantial portion of MSS cases, which is important for clinical application. Our trade-off analysis further demonstrates the model's ability to adapt thresholds for prediction. For instance, by lowering the false negative percentage to 0.05%, the sensitivity increases to above 99%, while still correctly classifying over 44% of the total patient cohort as MSS. These results highlight the flexibility of our model's adjustable threshold, enabling us to optimize performance metrics to suit a variety of clinical situations and user needs.

Additionally, our study has harnessed the interpretative power of attention heatmaps (Fig. 2e and Supplementary Figs. 2–4), which serve as a visual guide to the model's decision-making process. These heatmaps provide valuable insights that can assist pathologists in verifying known biomarker associations and potentially uncovering novel patterns. To further leverage these insights, we plan to collaborate with expert pathologists in future studies to systematically evaluate the highlighted regions and refine the model's interpretability, ensuring its clinical relevance and applicability.

### Comparative analysis of H&E and IHC inputs in PD-L1 expression AI models for BRCA

Pembrolizumab has received approval for treating patients with TNBC with a PD-L1 Combined Positive Score (CPS) of 10 or higher[2,49]. Unlike the straightforward evaluation of dichotomous IHC staining, the assessment of PD-L1 status is complicated by an additional scoring system, introducing greater complexity and variability. The prediction of PD-L1 expression using AI tools has become a research topic of great interest[27,50–52]. With our dual-modality model, we have achieved AUROC scores of 0.866 for H&E-only inputs, 0.959 for IHC-only inputs, and 0.957 for combined H&E/IHC inputs (Fig. 3a).

Performing stratified analysis, we observed that the H&E-only model exhibited better performance with primary site samples, and the type of scanner utilized had a negligible influence on the model's effectiveness. In contrast, for IHC-stained cases, neither the specimen site type nor the scanner type stratification affected the model's performance (Fig. 3b). Additionally, a trade-off analysis was conducted to demonstrate the model's versatile applicability in various clinical settings (Fig. 3c and Supplementary Data 3). Attention and classification heatmaps were also generated for interpretation (Fig. 3d and Supplementary Figs. 5 and 6). The histograms representing the distribution of prediction probabilities are depicted in Supplementary Fig. 7. Our observations indicate a more discernible separation of prediction probabilities when using the IHC or H&E/IHC-based models, as opposed to the model based solely on H&E.

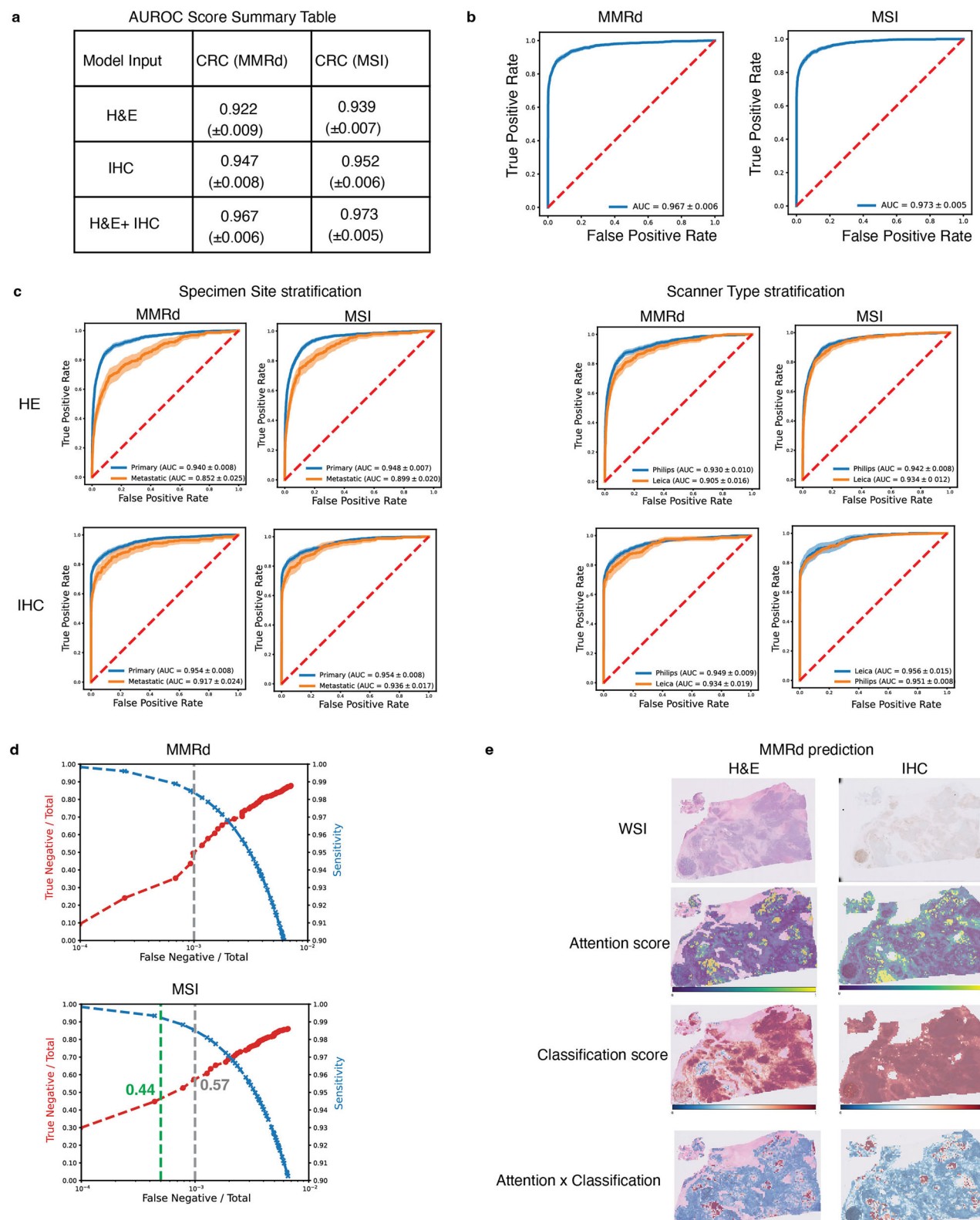

**a** AUROC Score Summary Table

| Model Input | CRC (MMRd) | CRC (MSI) |
|---|---|---|
| H&E | 0.922 (±0.009) | 0.939 (±0.007) |
| IHC | 0.947 (±0.008) | 0.952 (±0.006) |
| H&E+ IHC | 0.967 (±0.006) | 0.973 (±0.005) |

## Prognostic significance of predicted biomarkers in pembrolizumab treatment outcomes

In order to elucidate the prognostic impact of our biomarker predictions on clinical outcomes, we conducted survival analyses utilizing two distinct endpoints: time-on-treatment (TOT) and overall survival (OS) (Supplementary Tables 7 and 8). In the cohort of patients with CRC treated with

pembrolizumab, those exhibiting MSI-H or MMRd showed notably longer TOT and OS. Specifically, patients with MMRd had a hazard ratio (HR) of 0.463 (95% confidence interval [CI]: 0.365–0.587, $p < 0.001$) for TOT and of 0.398 (95% CI: 0.282–0.562, $p < 0.001$) for OS. These trends were consistently observed with our predictive models for MSI and MMRd status. For example, when performing survival analysis utilizing MMRd status

**Fig. 2 | Assessment of biomarker prediction performance for MSI and MMRd in CRC. a** Aggregated area under the receiver operating characteristic (AUROC) scores for predictions of microsatellite instability-high (MSI-H) and mismatch repair deficiency (MMRd) across three model types: hematoxylin & eosin (H&E) only, immunohistochemistry (IHC) only, H&E/IHC duet. Each value is the mean from a five-fold cross-validation ± 95% confidence interval (95% CI). **b** AUROC curves for MMRd and MSI prediction performance of the H&E/IHC duet model. **c** Stratified analysis of MMRd and MSI predictions, differentiated by specimen site and scanner type. **d** Cost-benefit analysis of duet model predictions. The X-axis indicates the False Negative Percentage (False Negatives/Total Cases). The left Y-axis represents the True Negative Percentage (True Negatives/Total Cases), and the right Y-axis denotes Sensitivity. **e** Visualization of attention and classification scores for deficient MMR specimens. The attention heatmap illustrates the per-patch attention rollout of our trained transformer-based feature aggregation duet model, with larger values (yellow) indicating higher contribution to the model's prediction and smaller values (purple) indicating lower contribution. The classification heatmap displays the per-patch MMRd classification scores, with deficient MMR as the positive class and proficient MMR as the negative class. The attention x classification heatmap highlighted tiles that provide final weighted classification score.

predicted by the duet model, the HR for TOT was 0.506 (95% CI: 0.4–0.641, $p < 0.001$) and for OS was 0.38 (95% CI: 0.267–0.539, $p < 0.001$), aligning closely with the actual status trends. Comparing all three models, the MMRd status predicted by the duet model demonstrated stronger association with clinical outcomes than the other two single-stain WSI models (Fig. 4a–d). Similar patterns were observed when performing survival analysis based on MSI status (Fig. 5a–d).

Conversely, among patients with BRCA treated with pembrolizumab, extended TOT and OS were noted among patients with a CPS of 10 or above, but only TOT reached statistical significance. Specifically, patients with CPS of 10 or above had a HR for TOT of 0.785 (95% CI: 0.629–0.979, $p < 0.05$) and for OS of 0.882 (95% CI: 0.626–1.241, $p > 0.1$). Intriguingly, when we applied predicted CPS statuses for clinical outcome analyses, the prognoses inferred from the H&E model surpassed the actual pathologist-scored CPS in significance. The HR for TOT was 0.671 (95% CI: 0.525–0.858, $p < 0.005$) and for OS was 0.511 (95% CI: 0.358–0.729, $p < 0.001$) (Fig. 6).

## Discussion

The development of advanced algorithms and the improvement of computing hardware have given researchers powerful tools to apply AI in healthcare[53,54]. AI models have been developed to tackle complex challenges such as predicting biomarker status, clinical outcomes, and classifying cancer subtypes[55–59]. Recent perspectives emphasized that AI-based biomarkers derived from routine clinical data could greatly improve the accessibility of personalized medicine by providing rapid, cost-effective alternatives to traditional molecular testing. In addition, AI-powered decision support systems have the potential to reduce the workload of healthcare practitioners, particularly in smaller oncological centers with limited resources, by automating time-consuming tasks and streamlining patient stratification[58]. In our study, we demonstrate a dual-modality approach that significantly increases the accuracy of MSI and MMRd predictions compared to single-input models and enhances their correlation with clinical outcomes. This research represents a notable step forward in employing AI to analyze oncological biomarkers, highlighting its substantial potential to improve patient treatment outcomes.

Immunotherapy has become a cornerstone of treatment for multiple cancer types[60]. In this study, we trained and evaluated our model using a substantial cohort, encompassing over 20,000 patients with CRC and more than 15,000 patients with BRCA. Patient specimens were sourced from a variety of clinics and research facilities, with tissue samples ranging from primary to metastatic sites. Collection methods were diverse, including resection, fine needle biopsy, core needle biopsy, and others. Additionally, the pathological slides were processed using two distinct scanners from Philips and Leica, which underscores the heterogeneity of our data set. This diversity in data sources and processing methods highlights the strong adaptability and robustness of our model, as it is trained on a wide array of samples, ensuring better generalization to real-world clinical scenarios.

Undoubtedly, our cohort reflects the intricate diversity present in tumor specimens from patients, which is essential for developing clinically relevant models. Despite the inherent variability, our model demonstrated remarkable predictive accuracy for biomarkers, highlighting its potential clinical value. Our stratified analysis revealed that the conditions of WSI—specifically, the site of the specimen and the type of scanner used—

contributed differentially to model performance. This not only underscores the model's robustness but also its adaptability to diverse clinical conditions.

Our research further extends beyond mere biomarker prediction. We aimed to understand how predictions from our model correlate with clinical outcomes. Our findings indicate that the AI-predicted biomarker status correlates with clinical outcomes as strongly as—and in some cases more strongly than—biomarker status annotated by pathologists. This holds promise for integrating AI into clinical decision-making processes. For example, PD-L1 status, as predicted by our H&E WSI-based model, correlated with patient outcomes more effectively than the CPS scores calculated by pathologists. The precision of our model's predictions may pave the way for refining patient selection criteria for pembrolizumab therapy. Implementation of our models could add an additional layer of precision to existing practices. This collaborative model between AI and human expertise could set a new precedent in the field, pending successful validation and adoption as a standard of care.

Our methodological approach, which integrates H&E and IHC images via a dual-modality transformer-based pipeline, represents, to the best of our knowledge, a novel strategy in medical AI. To further validate our approach, we performed additional experiments using several newly released pathology foundation models—Virchow[61], Virchow2[62], UNI[63], and H-Optimus-0[64]—as feature encoders. Specifically, we extracted feature embeddings from both H&E and IHC images and then fed these embeddings into either our duet model or a single-staining model. The results, summarized in Supplementary Table 9, demonstrate that our duet model consistently outperforms the single-staining model for most biomarkers, confirming the value of integrating morphological and immunohistochemical information.

Interestingly, for BRCA–PD-L1 prediction, we observed no meaningful increase in predictive performance when adding H&E data to the IHC-only model. This finding suggests that PD-L1 status in BRCA can often be captured effectively through a single immunostain, thereby diminishing the contribution of additional morphological cues. In contrast, we saw marked improvements in performance for CRC–MMRd and CRC–MSI predictions when combining H&E and IHC data. Because our analysis used only MLH1 IHC among the four MMR proteins, the morphological features in H&E slides likely complemented the single IHC stain, capturing a broader range of tumor characteristics that boosted the model's accuracy. We also noted that using H&E alone produced an AUROC below 0.9 for PD-L1 prediction in breast cancer but exceeding 0.9 for MMRd/MSI prediction in CRC, underscoring that the magnitude of histopathological changes associated with a given biomarker may determine how much H&E data contribute to predictive power.

These observations highlight two important points. First, our duet model shows clear advantages for biomarkers that exhibit strong morphological correlates—such as MMRd or MSI—where the synergy between H&E and IHC data appears to be most beneficial. Second, not all biomarkers may benefit equally from dual-staining approaches. For markers like PD-L1 in BRCA, which may be adequately characterized by a single IHC stain, adding H&E data may not contribute additional benefits. Overall, these experiments strengthen our claim that combining H&E and IHC images can provide superior predictive performance in many contexts. The choice between single- or dual-staining strategies may ultimately depend on the biomarker in question, the nature of its morphological footprint, and the specific research or clinical objectives.

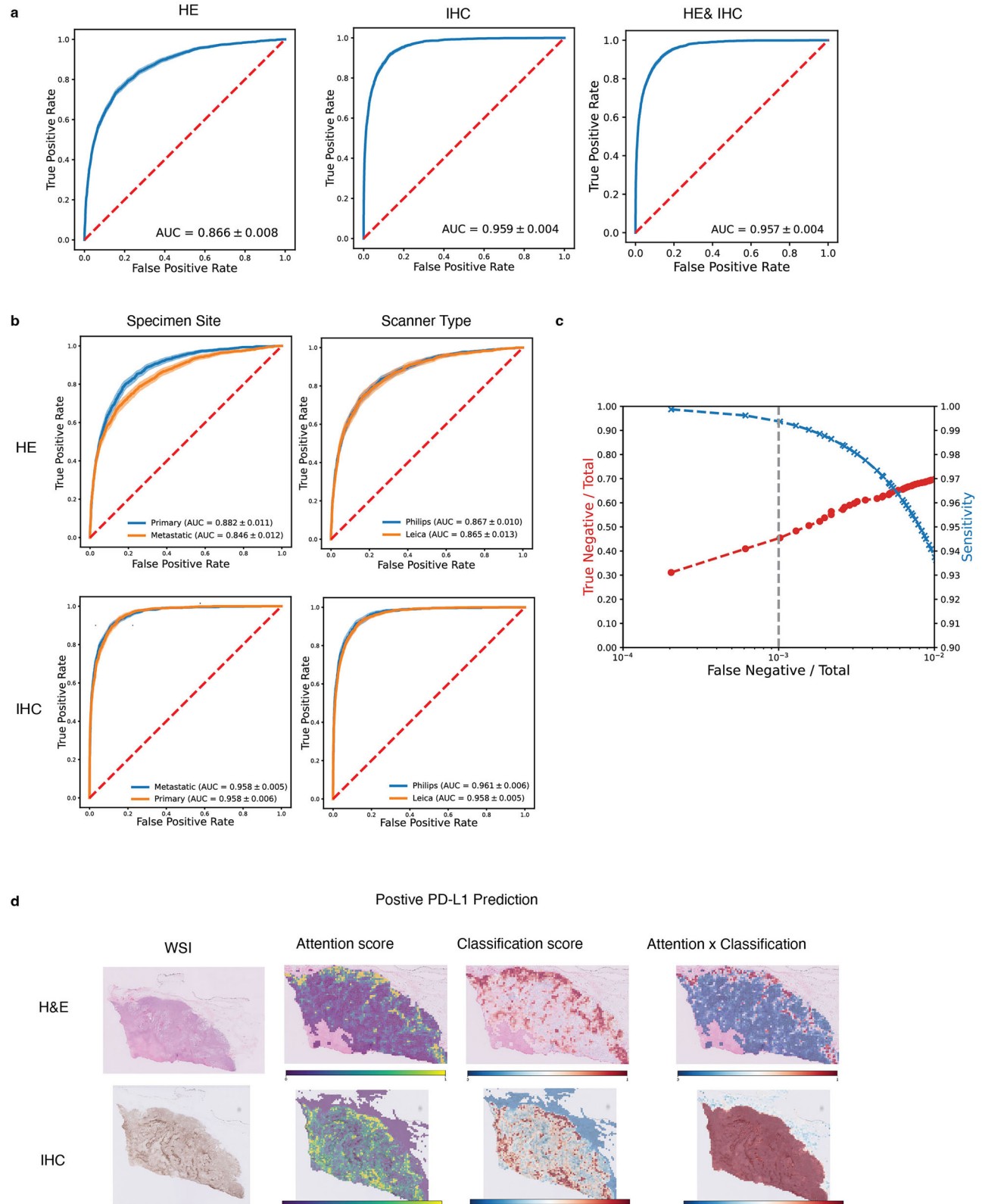

**Fig. 3 | Evaluating biomarker prediction for PD-L1 status in BRCA. a** Area under the receiver operating characteristic (AUROC) curves for the prediction of PD-L1 status (where CPS ≥ 10 denotes PD-L1 positivity), comparing the performance of the hematoxylin & eosin (H&E) only, immunohistochemistry (IHC) only, and H&E/IHC duet models. **b** Stratified analysis of PD-L1 status predictions by specimen site and scanner type, elucidating the model's performance across different conditions. **c** Cost-benefit analysis of the duet model's predictions, with the False Negative Percentage (False Negatives/Total Cases) on the *X*-axis, the True Negative

Percentage (True Negatives/Total Cases) on the left *Y*-axis, and Sensitivity on the right *Y*-axis. **d** Visualization of attention and classification scores for PD-L1-positive specimens. The attention heatmap conveys the per-patch significance through our trained transformer-based feature aggregation duet model, where yellow indicates a high contribution and purple a low contribution to the model's output. The classification heatmap portrays the per-patch PD-L1 classification scores. The attention x classification heatmap highlighted tiles that provide final weighted classification score.

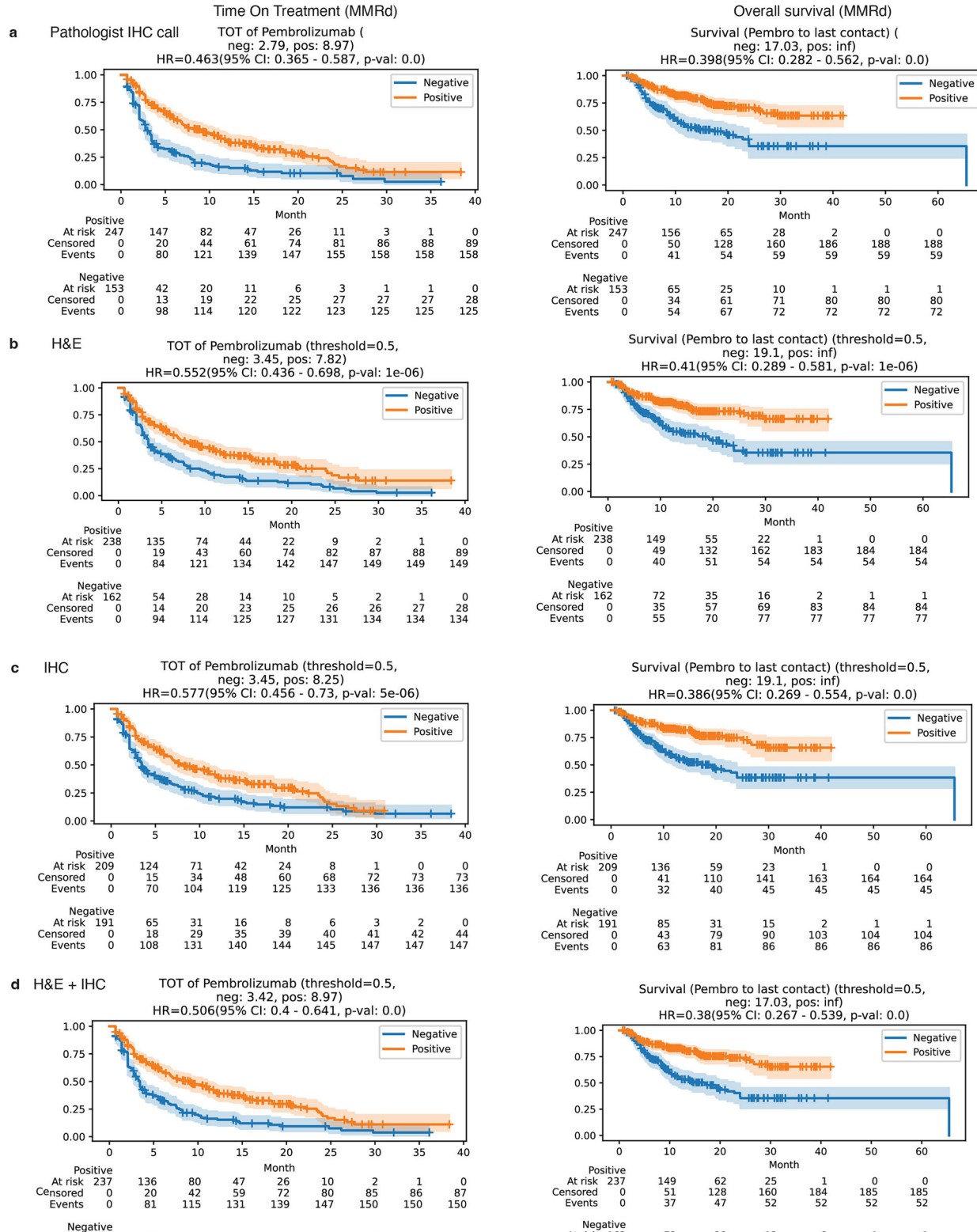

**Fig. 4 | Comparative survival analysis: evaluating the impact of ground truth and predicted MMRd status in CRC patients.** Kaplan–Meier survival curves for patients categorized by mismatch repair deficiency (MMRd) status, determined through pathological assessment (**a**), prediction from hematoxylin & eosin (H&E) whole slide images (WSIs) (**b**), prediction from immunohistochemistry (IHC) WSI (**c**), and prediction from a combined H&E/IHC WSI approach (**d**). The hazard ratio (HR) for the MMRd group is provided, with the mismatch repair proficient (MMRp) group serving as the reference. The shaded regions indicate 95% confidence intervals (CI). The p values were derived using the log-rank test to compare each MMRd group with the respective MMRp group.

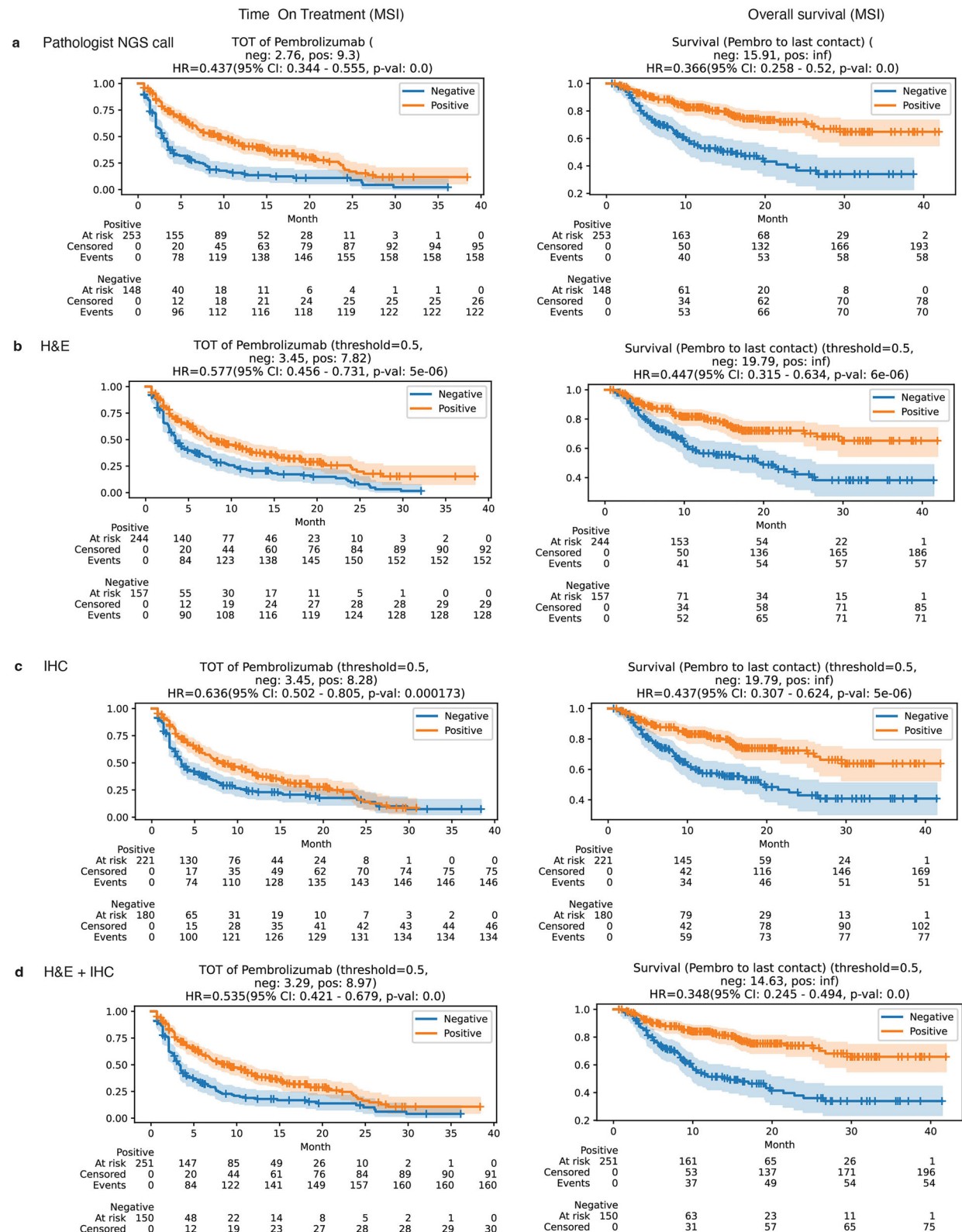

**Fig. 5 | Comparative survival analysis: evaluating the impact of ground truth and predicted MSI status in CRC patients.** Kaplan–Meier survival curves for patients classified by MSI status, determined through pathological assessment (**a**), prediction from H&E WSI (**b**), prediction from IHC WSI (**c**), and prediction from a combined H&E/IHC WSI method (**d**). The HR for the MSI group is provided, using the microsatellite stable (MSS) group as the reference. Shaded areas delineate 95% confidence intervals. *p* values were calculated using the log-rank test to contrast each MSI group with the corresponding MSS group.

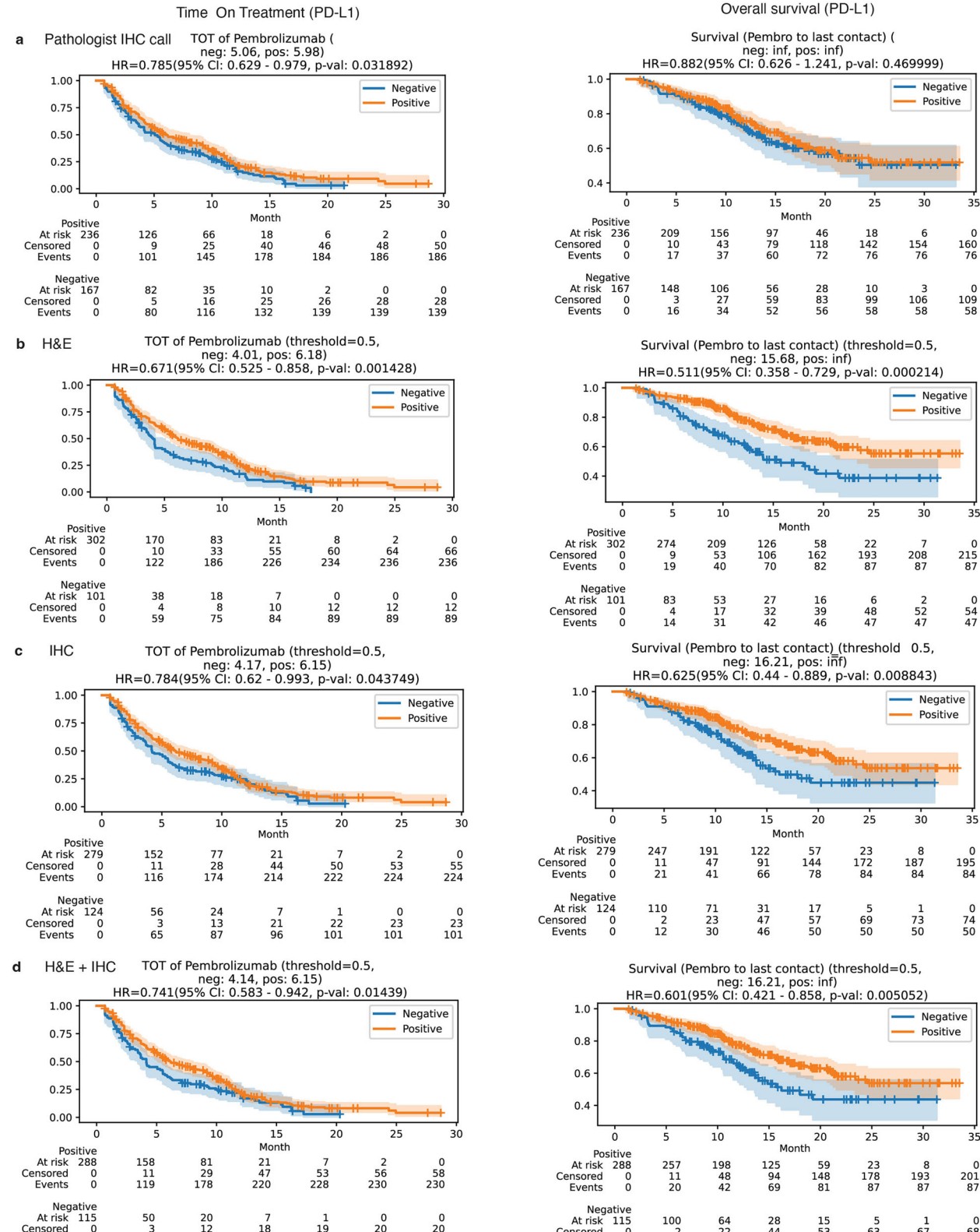

**Fig. 6 | Comparative survival analysis: evaluating the influence of actual and predicted PD-L1 status in patients with BRCA.** Kaplan–Meier curves comparing patient groups classified by PD-L1 status, which is determined through different methods: pathological evaluation (**a**), prediction using hematoxylin & eosin (H&E) whole slide images (WSIs) (**b**), prediction using immunohistochemistry (IHC) WSI (**c**), and prediction using a combined H&E/IHC WSI model (**d**). The hazard ratio (HR) is reported for the PD-L1-positive group with the PD-L1-negative group as the baseline for comparison. Confidence intervals (CI) at 95% are shown as shaded regions around the curves. We computed *p* values by employing the log-rank test to compare the survival rates of each PD-L1-positive group against their PD-L1-negative counterparts.

Beyond our exploration of various feature encoders, we see additional opportunities to expand our framework into a fully multimodal system. Our model can also be further refined to be a multimodal system capable of incorporating multiple staining features, which could be invaluable for subtyping cancers, such as BRCA, by concurrently assessing ER, PR, and HER2 IHC staining. Furthermore, the incorporation of genomic data into our model represents our commitment to enriching the model's learning with a multidimensional dataset. Recent reviews by Unger et al.[65,66] summarized current and emerging applications of deep learning in histopathology and genomics. Several studies have adopted multimodal models to use several data types simultaneously, such as combining histopathological images with genetic data[67–71]. Additionally, researchers have also utilized radiology images in addition to histologic and genomic data, which adds another layer of rich information[72,73]. Moreover, a recent study introduced a multistain deep learning technique to integrate multiple IHC tissue microarray images to enhance the accuracy of prognosis and therapeutic response predictions in CRC[74]. In conclusion, the accumulated evidence from various studies reinforces the premise that multimodal approaches, which harness the synergistic potential of multiple data types, can outperform analyses that depend on a single modality. By integrating diverse data sources that offer a more comprehensive view of patient-specific characteristics, we will be able to tailor treatment to individual patient profiles.

Our study also highlights the practical application of our model's predictions through trade-off analysis, aiming to evaluate the efficiency of AI-based tools in clinical settings, such as the pre-screening of patients using routine H&E or combined H&E/IHC without pathological scoring. The previously reported MSIntuit tool demonstrates notable efficacy, ruling out almost half of the non-MSI population while accurately classifying over 96% of MSI patients, surpassing the current gold-standard methods (92–95%)[46]. In comparison, our H&E WSI-based model ruled out 42.8% of the non-MSI population at a sensitivity of 98% and 62.4% at a sensitivity of 96% (Supplementary Data 4). Our DuoHistoNet model further improved performance, ruling out 65.1% of the non-MSI population at a sensitivity of 98% and 78.3% at a sensitivity of 96% (Supplementary Data 2). This indicates that our dual-modality approach provides pathology laboratories with a more effective tool for MSI pre-screening. It achieves higher specificity than MSIntuit while maintaining high sensitivity.

We must acknowledge that biomarker testing is time-intensive and that pathological scoring systems, such as CPS, are not only complex but also subject to inter-pathologist variability[75]. Our duet system can assist in mitigating these issues. We believe our methodology could aid oncologists in making informed decisions and expedite the delivery of optimal treatments to patients. Additionally, it could help reduce costs for pathology labs. In line with the suggestions of Kacew et al.[76], we plan to collaborate with pathology labs and hospitals to conduct a medico-economic evaluation of AI-enabled solutions, with the goal of confirming potential cost reductions in cancer management.

Incorporating heatmap visualizations has enhanced our ability to pinpoint key morphological features that our model deems critical for predictions. These high-ranking tiles highlight the intricate visual patterns that the model associates with specific molecular alterations, offering a novel lens through which to examine histomorphology. Future collaborations with pathologists will be crucial for systematically evaluating and classifying these heatmaps, helping to validate the morphological signals the model relies upon and to investigate whether previously unlinked visual features may have clinical relevance. Examining correctly classified cases is essential to ensure that the model's predictions align with biologically meaningful morphological features rather than spurious patterns. Additionally, a more comprehensive review of misclassified cases could help identify potential confounding factors that challenge model performance. By bridging AI-driven analyses with expert pathological insight, we hope to refine our approach further, potentially uncovering new morphological indicators of tumor heterogeneity or treatment response. Through ongoing validation efforts, we aim to strengthen the clinical utility of these heatmaps and ultimately improve patient care.

A key limitation of this study is that MMRd was assessed using only MLH1 IHC, without inclusion of PMS2, MSH2, or MSH6 staining. Although our dual-modality model achieved a high AUROC of 0.967, relying on a single IHC marker may lead to misclassification in cases where MLH1 expression is retained despite isolated loss of other MMR proteins. Comprehensive MMR assessment typically requires evaluation of all four markers to ensure accurate identification of MMRd tumors. Pearlman et al.[77] showed that two-stain strategies using MSH6 and PMS2 can fail to detect a subset of Lynch syndrome cases in which MSH2 is absent but MSH6 is preserved or equivocal, underscoring the need for the full four-marker panel. MLH1 loss, often resulting from promoter hypermethylation, is the most common cause of MMR deficiency in sporadic colorectal cancer[48] and thus captures a large proportion of MMRd cases. Additionally, the integration of H&E images in our model may help recover some of the morphologic patterns associated with MMRd, potentially reducing the impact of missing IHC markers. However, this cannot fully substitute for direct assessment of the complete MMR protein panel. Future studies will incorporate PMS2, MSH2, and MSH6 staining to enable more comprehensive MMRd classification, minimize potential misclassification, and enhance the clinical utility of the model.

Lastly, while our model was trained and validated on a sizable and varied dataset, and further analysis of stratification underscored its robustness and adaptability, we acknowledge the importance of external validation. Testing our model on an independent dataset would be ideal; however, there currently appears to be no publicly available dataset that includes both H&E and IHC images for such use. We are actively seeking partnerships for external validation to enhance the credibility of our model. Moreover, the reliability of AI-generated predictions is fundamentally linked to the quality of the input data. There is a crucial need for continuous monitoring to address and reduce the possibility of biases that could impact the model's performance and, by extension, its clinical applicability. Looking forward, we anticipate the deployment of our model as an adjunct to pathologist-led biomarker evaluation. Continuous validation against novel datasets remains a priority, as does the expansion of our model to include other emerging biomarkers.

In conclusion, our study introduces a framework that transcends conventional pathology slide analysis. We have demonstrated the capability of our model to predict critical prognostic biomarkers for immunotherapy, highlighting its potential as a supportive tool for pathologists. The heatmaps generated by our model not only facilitate the identification of impactful patches or regions of interest but also foster a collaborative environment where pathologists can interpret AI findings. This synergy could uncover new diagnostic patterns and paradigms, potentially revolutionizing patient stratification and prognosis. Ultimately, we envision AI tools delivering multifaceted support to contemporary and future clinical practices.

## Data availability

The data generated in this study are not publicly available due to patient privacy but will be made available for replication and verification purposes from the corresponding author upon signing of a data usage agreement.

## Code availability

The code was developed based on the open-source project HistoBistro (https://github.com/peng-lab/HistoBistro). The code will be made available for replication and verification purposes from the corresponding author upon signing a code usage agreement and subject to company approval.

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

## Acknowledgements

We would like to express our gratitude to Sarah Sprinkle, Josh Lawrimore, and Vaishali Pannu for their invaluable assistance in identifying and collecting additional samples for our study. We are also thankful to Josh Lawrimore, Barry Weickert, and Sergey Klimov for their insightful exchanges and contributions to our discussions throughout the research process.

## Author contributions

Study conception and design: G.W.S. and M.C.; data collection and processing: Y.C., N.L., M.C., L.D., and M.R.; analysis and interpretation of results: Y.C., N.L., M.C., E.A., M.A.R., J.X., A.H., L.R., and G.W.S.; draft manuscript preparation: Y.C., N.L., M.C., J.X., J.R.R., H.G., M.O., D.S., and G.W.S. All authors reviewed the results and approved the final version of the manuscript.

## Competing interests

Y.C., N.L., M.C., E.A., M.R., M.A.R., J.X., A.H., L.D., J.R.R., H.G., M.O., D.S., and G.W.S. have a financial relationship as employees of Caris Life Sciences. The authors declare that they have no other known competing financial interests or personal relationships that could have appeared to influence the work reported in this paper.
