## [Transparent Peer Review file · Communications Medicine]

Synergistic H&E and IHC Image Analysis by AI Predicts Cancer Biomarkers and Survival Outcomes in Colorectal and Breast Cancer

Corresponding Author: Dr George W Sledge

Version 0:

Reviewer comments:

Reviewer #1

(Remarks to the Author)

The paper on "Dual-Modality AI Framework: Synergistic H&E and IHC Image Analysis for Predicting Cancer Biomarkers and Survival Outcomes in Colorectal and Breast Cancer" proposes a novel, dual-modality transformer-based model for predicting MSI/MMRd and PD-L1 status using hematoxylin & eosin (H&E) and IHC stained whole slide images. The paper is well written and results are quite promising. However, following are some of my suggestions to be incorporated to justify the superiority of the proposed systems:

1. Provide a comparison of your proposed methods with other existing AI models for the prediction of Cancer biomarkers and survival outcomes, in order to justify that the proposed model is superior to the existing techniques.
2. For the better readability of the paper, include a more detailed literature review.

Reviewer #2

(Remarks to the Author)

Cheng and Lama et al. perform a study using large internal cohorts of CRC MMR, MSI and BRCA PD-L1 cases with paired HE and IHC stained WSIs to develop weakly-supervised deep learning-based biomarker prediction models. Each model takes either H&E, IHC or both as input (after suitable preprocessing and featurization using a pretrained encoder), and outputs classification predictions for the given marker. Overall impressive performance is demonstrated, especially when using IHC or when both IHC / HE are combined, with the models reaching ~0.95 AUC for IHC and ~0.97 AUC for IHC+HE on the CRC cohorts and ~0.96 on the BRCA cohort. KM-survival analyses were further performed to demonstrate statistically significant stratification of patients based on predicted biomarker status. However, in its current form, several limitations exist that should ideally be addressed.

1. How exactly is branch dropout applied? Since the output features of each branch are concatenated together before feeding to the classifier, does dropout in this case mean that all features from a given branch are "zeroed" out? Is this what it means to apply the model to a single modality (i.e. HE or IHC alone) at test time as well?
2. In the methods, it mentions: "Upon completion of training, we evaluated the model's performance across each fold's test set as well as on an external holdout dataset", however, unless I missed it, no such external holdout dataset is introduced for additional validation. Please clarify.
3. There is no distribution provided for relevant attributes such as the number of positive / negative samples for each cohort, how many samples are scanned with each scanner type, how many are primary vs. metastatic tumors.
4. There is currently no ablation experiments to provide any valuable insight as to why the model performs well and how the different choices of the model and training design influence performance. For example, why is CTransPath chosen as the image encoder when it was only pretrained on H&E images? Would the authors benefit from choosing a different image encoder that was pretrained with both H&E and IHC images? How is the branch dropout rate / learning rate etc. chosen?
5. Since the model and data cannot be made available, it would be very helpful if the authors can also benchmark on a publicly available dataset for not only external validation, but also to allow comparisons with the broader literature of works that study e.g. MSI prediction using deep learning. The authors claim that the lack of external validation is because there are

no publicly available datasets with both IHC and H&E slides - but since the model works also for H&E alone, I don't see why it cannot be evaluated on publicly available H&E only datasets such as TCGA CRC.

6. I am confused by the sensitivity / false negative rate calculation on pg 8, 1158 - 167. How is it possible that the sensitivity increases from 98%  0.99% while false negative rate is reduced from 0.1% to 0.05%? What is the meaning of "This allowed for the accurate detection of 57% of Microsatellite Stable (MSS) patients", is this referring to the NPV?

Reviewer #3

(Remarks to the Author)

Overall, this is a well-written manuscript and a nice study, which tackles a timely and important topic. The dataset is diverse and huge, the pipeline robust and well-designed. However, there are some questions/points that need to be addressed in a revised version.

Major:

- The authors should clarify already in the results section which IHC stains were used (MLH1/PMS2/MSH2/MSH6?).
- Innocently provocative point: If MMR stains are available, why should we "predict" MMR status? For a pathologist, it takes not much time to analyze such an IHC. Do the authors have any idea how their approach deals with difficult and/rare staining patterns such as described here DOI: 10.1038/s41379-022-01109-4, especially clonal loss/heterogeneous staining (please also include this in the methods section how these rare staining patterns were classified, lines 365-369). Moreover, it would be highly interesting to look which cases were mis-classified by the model even though it had H&E as well as IHC data. As a pathologist, I would be quite interested to look into those cases more closely and think this is also of broader interest. Maybe the authors could make these cases publicly available with their publication and/or include representative cases in their supplement. A pathologic review of the misclassified cases should be performed.
- What did the authors see in their heatmaps (lines 169-172)? Was a pathological review performed? Any new insights? As they describe that this has the potential to identify new patterns? In Supplementary Figure 1a the H&E shows an adenocarcinoma with gland-forming morphology (NOS) and retained MMR staining (which MMR marker IHC is displayed here?). In Supplementary Figure 1b a potentially medullary CRC is displayed which is a typical MSI-like morphology. Moreover, the normal mucosa shows retained MMR expression and is not highlighted in the heatmap. Comparing MSI prediction heatmaps and IHC stains, especially in heterogeneous cases, could be highly interesting and could prove that DL-based MSI predictors can resolve intratumoral heterogeneity on a biologic basis. For references on these topics: doi: 10.1038/modpathol.2016.198, doi: 10.1136/gutjnl-2019-319866)
- Lines 127-128: MLH1 promoter hypermethylation and consecutive loss of MLH1 expression is the most common reason for dMMR (doi: 10.1053/j.gastro.2009.12.064) but I think the phrase "due to its high concordance with MMR status" is a bit misleading. If a case shows MLH1 expression loss, it is per se dMMR. As MMR proteins function as heterodimers (MLH1/PMS2; MSH2/MSH6), other staining patterns can be observed – even the loss in both heterodimers (<https://doi.org/10.3389/fonc.2022.1019798>).
- How do the authors explain the poorer performance of MMR prediction when focusing on metastatic specimens (resection specimens and/or biopsies)? In the literature, it is known that performance drops when trying to predict MSI status from biopsy material and/or metastases. However, when integrating IHC this should be improved. Do the authors have further ideas how MSI prediction on biopsies and/or metastatic sites could be improved in the future, as these two indications are very important: neoadjuvant IO-therapy (NICHE2, DOI: 10.1056/NEJMoa2400634) and IO-therapy in the palliative/metastatic setting.
- The authors argue that their model can reduce the necessity for NGS-based MSI testing. This is correct. However, this is also correct for a solely HE-based MSI deep-learning predictor such as MSIntuit by Owkin (in this scenario you even save money and time for IHC). Moreover, by performing (large panel) NGS testing that allows MSI assessment you gain further important information on driver mutations such as KRAS, BRAF etc. but also on rare but important mutations in CRC such as POLE. Could the authors please comment on this or even include their thoughts into the manuscript. Could the authors also bring up arguments why their pipeline is superior to e.g. MSIntuit – if they believe this is the case.
- It is great that the authors include survival analysis. However, if survival data is available, the authors could try to predict survival and/or treatment response directly from HE and IHC, and without the necessity/detour of predicting MSI/dMMR status.
- The HE-model in breast cancer surpassed the actual pathologist-scored CPS in significance. What are the potential reasons for this? This is interesting. To the authors think this has a biologic background. Were heatmaps revisited? Were the actual pathologist scores re-viewed if they were correct?

Minor:

- Line 51-52: MSI status assessment most often relies on PCR-based methods, but NGS is becoming more prevalent. For MMR testing, IHC is of course correct.
- Lines 61-67: For an overview of DL-based techniques in CRC refer to the extensive literature on this, e.g. doi: 10.1159/000539678.
- Patient Characteristics: I assume that the CRC MMRd cohort was tested via IHC and the CRC MSI cohort via PCR? But could you please clarify. If possible, please provide more detailed clinicopathological characteristics of the cohorts (pTNM stage, L/V/Pn, Budding, hormone receptors/HER2/Ki67 status, if available). CRC as well as BRCA are really heterogeneous diseases.
- Why is the model framework "revised"? Is there any prior version? If so, please cite/refer to this.
- Please tone down in lines 108-110: Other studies also use ViT for HE and IHC analysis (just as example

<https://doi.org/10.1016/j.media.2024.103289>). Of course, the field is here rapidly evolving.

- Lines 121-123: Different algorithms have shown to predict MSI, however here only the Wagner et al. paper on transformer-based MSI prediction is cited (in Wagner et al. AttMIL, CNN etc are also included). However, when talking about different multiple studies the authors should provide adequate references.
- Lines 124-125: Is mentioning a CT-based study here necessary? If so, the unique approach with predicting MSI status not from tissue biopsies but non-invasively via imaging data should be made clear.
- The risk tables in the Kaplan Meier curves are quite small.

Version 1:

Reviewer comments:

Reviewer #2

(Remarks to the Author)

Thank you for updating the manuscript with additional experiments and clarification - they have addressed all my concerns.

Reviewer #3

(Remarks to the Author)

I appreciated the extensive work by the authors done in the revision.

As the authors only used MLH1-staining for MMR-status, this should be stated as major limitation. They also state that this may lead to misclassification (retained MLH1-expression leading to pMMR prediction etc) despite of immunohistochemical loss of PMS2, or MSH2 or MSH6. Even in two stain approaches, pathologists use PM2/MSH6 staining (<https://pubmed.ncbi.nlm.nih.gov/29967423/>).

The authors should do follow-up studies with all immunohistochemical markers.

We thank the reviewers for their constructive comments and suggestions. We have carefully considered each point raised and have revised the manuscript accordingly. Below, we provide a point-by-point response to all reviewer comments, detailing the changes made to enhance the quality and clarity of our work. Changes are highlighted in yellow in the tracked changes version of the document. We believe these revisions have significantly strengthened our manuscript.

Reviewer #1 Comments and Responses

Reviewer #1 (Remarks to the Author):

The paper on “Dual-Modality AI Framework: Synergistic H&E and IHC Image Analysis for Predicting Cancer Biomarkers and Survival Outcomes in Colorectal and Breast Cancer” proposes a novel, dual-modality transformer-based model for predicting MSI/MMRd and PD-L1 status using hematoxylin & eosin (H&E) and IHC stained whole slide images. The paper is well written, and results are quite promising. However, following are some of my suggestions to be incorporated to justify the superiority of the proposed systems:

- 1. Provide a comparison of your proposed methods with other existing AI models for the prediction of Cancer biomarkers and survival outcomes, in order to justify that the proposed model is superior to the existing techniques.**

Response: We appreciate the reviewer’s suggestion. At the time of our project, the method proposed by Wagner et al. (Wagner, Reisenbuchler et al. 2023) was considered state-of-the-art for biomarker prediction. Our single-input model closely parallels their approach, with the only difference being our tile preprocessing step—Wagner et al.’s work did not include tissue segmentation or tile processing code.

Since Wagner et al.'s study had already compared their Transformer model to various existing AI methods and demonstrated superior performance, we did not undertake a separate comparison. Notably, our proposed duet model shows a significant improvement over the single-input H&E approach in predicting MSI/MMRd biomarkers.

Further, in the revised manuscript, we performed additional experiments using several newly released pathology foundation models—Virchow, Virchow2, UNI, and H-Optimus-0—as feature encoders for the histopathology images. Specifically, we utilized these encoders to (1) generate feature embeddings from the images and (2) input those embeddings into both our duet (dual-staining) model and a single-staining model. We then evaluated each configuration by measuring the AUROC for various biomarkers.

As summarized in **Supplementary Table 9**, while the overall AUROC values depend on which foundation model is employed, our duet model consistently outperforms the single-staining model across most scenarios. The only exception is BRCA-PD-L1 prediction, where we observed no significant AUROC improvement using the duet model compared to the single-staining (IHC-only) model. We hypothesize that this is because PD-L1 status can be effectively captured by a single IHC stain, limiting the contribution of additional H&E information. Conversely, for CRC-MMR and CRC-MSI prediction, we only used the MLH1 IHC image among the four MMR proteins, so incorporating H&E data provided significant complementary information and markedly boosted the model's performance. Furthermore, we noted

that using H&E alone yields an AUROC below 0.9 (0.866–0.886) for BRCA-PD-L1 prediction but exceeds 0.9 for CRC-MMR or CRC-MSI prediction. This finding suggests that the extent of histopathological changes associated with the biomarker in question may determine how much the H&E stain contributes to the model's predictive power. Hence, for biomarkers that produce more pronounced morphological alterations—such as MSI and MMR—the duet model shows clear gains over single-staining approaches. We have included a detailed discussion of these results and our underlying rationale in the revised manuscript (Lines 284-316).

Overall, these additional experiments bolster our claim that the duet model can offer superior predictive performance, particularly for biomarkers that produce pronounced histopathological changes or require staining of more than one protein. In such scenarios, the combined information from H&E and single IHC marker yields richer morphological and molecular features, thereby enhancing the model's predictive accuracy.

References

Wagner, S.J., et al. Transformer-based biomarker prediction from colorectal cancer histology: A large-scale multicentric study. Cancer Cell 41, 1650-1661 e1654 (2023).

2. *For the better readability of the paper, include a more detailed literature review.*

Response: We appreciate the reviewer's suggestion. In the revised manuscript, we have added additional references and expanded our discussion of related studies where appropriate to improve the paper's overall readability.

Reviewer #2 Comments and Responses

Reviewer #2 (Remarks to the Author):

Cheng and Lama et al. perform a study using large internal cohorts of CRC MMR, MSI and BRCA PD-L1 cases with paired HE and IHC stained WSIs to develop weakly-supervised deep learning-based biomarker prediction models. Each model takes either H&E, IHC or both as input (after suitable preprocessing and featurization using a pretrained encoder), and outputs classification predictions for the given marker. Overall impressive performance is demonstrated, especially when using IHC or when both IHC / HE are combined, with the models reaching ~0.95 AUC for IHC and ~0.97 AUC for IHC+HE on the CRC cohorts and ~0.96 on the BRCA cohort. KM-survival analyses were further performed to demonstrate statistically significant stratification of patients based on predicted biomarker status. However, in its current form, several limitations exist that should ideally be addressed.

1. How exactly is branch dropout applied? Since the output features of each branch are concatenated together before feeding to the classifier, does dropout in this case mean that all features from a given branch are "zeroed" out? Is this what it means to apply the model to a single modality (i.e. HE or IHC alone) at test time as well?

Response: We implement branch dropout by completely removing (i.e., zeroing out) the features from one branch during training, prior to concatenating them with the other branch. We do this for two main reasons:

- **Balancing Feature Importance:** In many cases, IHC features are more informative than H&E features, potentially leading the model to rely heavily on

IHC. By zeroing out one branch at random, we encourage the network to learn complementary information from both modalities rather than overfitting to a single feature type.

- **Single-Modality Robustness:** We want the model to remain functional even if only one WSI modality is available at inference. Although our dual-modality model accepts both H&E and IHC images—and we found a branch dropout probability of 0.3 to be optimal—performance decreases slightly when only one modality is provided, compared to a dedicated single-modality model.

For a fair comparison, we used the same transformer backbone as Wagner et al.

(Wagner, Reisenbuchler et al. 2023) in both our single- and dual-modality models.

Therefore, the single-modality system described in the manuscript is not simply the dual-modality model with one input removed; it is a separate model trained specifically for that modality.

References

Wagner, S.J., et al. Transformer-based biomarker prediction from colorectal cancer histology: A large-scale multicentric study. Cancer Cell 41, 1650-1661 e1654 (2023).

2. *In the methods, it mentions: "Upon completion of training, we evaluated the model's performance across each fold's test set as well as on an external holdout dataset", however, unless I missed it, no such external holdout dataset is introduced for additional validation. Please clarify.*

Response: Thank you for pointing out the confusion. We do not have an external holdout dataset in this study. Instead, we used an internal holdout dataset that is separate from the data involved in the five-fold cross-validation splits. To clarify this,

we have updated the manuscript text (line 530). We appreciate your feedback and have taken steps to ensure the wording is now accurate.

3. *There is no distribution provided for relevant attributes such as the number of positive / negative samples for each cohort, how many samples are scanned with each scanner type, how many are primary vs. metastatic tumors.*

Response: Thank you for highlighting this gap. In the revised manuscript, **Table 1** includes the distribution of positive and negative samples for the CRC (MMRd), CRC (MSI), and BRCA (PD-L1) cohorts. Furthermore, we have prepared **Supplementary Tables 3–8**, which provide additional details on biomarker status, scanner type, and specimen site type distribution for both the modeling and holdout datasets. We have also updated the text (line 99, line 149-150) to explicitly reference these tables, ensuring that all relevant distributions are clearly presented and easily accessible.

4. *There are currently no ablation experiments to provide any valuable insight as to why the model performs well and how the different choices of the model and training design influence performance. For example, why is CTransPath chosen as the image encoder when it was only pretrained on H&E images? Would the authors benefit from choosing a different image encoder that was pretrained with both H&E and IHC images? How is the branch dropout rate / learning rate etc. chosen?*

Response: At the onset of our study in 2023, CTransPath stood out as a leading transformer-based encoder for histopathology and had been successfully applied in multiple published projects (Wagner, Reisenbuchler et al. 2023, Jiang, Yin et al. 2024). In addition, CTransPath’s training set (including PAIP data <http://wiseaip.org/paip/guide/introduction>) does contain a subset of IHC images.

However, we recognize the recent surge of foundation models in digital pathology and have accordingly conducted additional experiments using Virchow, Virchow2, UNI, and H-Optimus-0 as feature encoders for H&E and IHC images. In these experiments, we generated feature embeddings from each encoder and then evaluated both our duet (dual-staining) model and a single-staining model across various biomarkers. Results (**Supplementary Table 9**) show that while overall AUROCs vary based on the chosen foundation model, our duet approach consistently outperforms the single-staining model in most scenarios. Notably, for BRCA-PD-L1 prediction, the AUROC gains from the duet model are minimal compared to using IHC alone, likely because PD-L1 expression is captured sufficiently by the IHC stain. In contrast, for CRC-MMR and CRC-MSI, we only used an MLH1 IHC image among the four MMR proteins, so combining it with H&E yielded significantly higher performance. Furthermore, we noted that using H&E alone yields an AUROC below 0.9 (0.866–0.886) for BRCA-PD-L1 prediction but exceeds 0.9 for CRC-MMR or CRC-MSI prediction. This finding suggests that the extent of histopathological changes associated with the biomarker in question determines how much the H&E stain contributes to the model’s predictive power. Hence, for biomarkers that produce more pronounced morphological alterations—such as MSI and MMR—the duet model shows clear gains over single-staining approaches. Overall, these additional experiments bolster our claim that the duet model can offer superior predictive performance, particularly for biomarkers that produce pronounced histopathological changes or require staining of more than one protein. In such scenarios, the combined information from H&E and single IHC

marker yields richer morphological and molecular features, thereby enhancing the model's predictive accuracy.

Regarding hyperparameter tuning, we chose branch dropout rate, learning rate, and other parameters via 5-fold cross-validation. We set the dropout probability at 0.3 after observing optimal performance on dual-modality data, and we adopted a 1×10^{-5} learning rate for the final models.

References:

Wagner SJ, *et al.* Transformer-based biomarker prediction from colorectal cancer histology: A large-scale multicentric study. *Cancer Cell* **41**, 1650-1661 e1654 (2023).

Jiang R, *et al.* A transformer-based weakly supervised computational pathology method for clinical-grade diagnosis and molecular marker discovery of gliomas. *Nature Machine Intelligence* **6**, 876-891 (2024).

5. *Since the model and data cannot be made available, it would be very helpful if the authors can also benchmark on a publicly available dataset for not only external validation, but also to allow comparisons with the broader literature of works that study e.g. MSI prediction using deep learning. The authors claim that the lack of external validation is because there are no publicly available datasets with both IHC and H&E slides - but since the model works also for H&E alone, I don't see why it cannot be evaluated on publicly available H&E only datasets such as TCGA CRC.*

Response: We appreciate the reviewer's suggestion. Our main objective is to demonstrate the benefits of a dual-modality framework (DuoHistoNet) over single-

stain approaches (H&E-only or IHC-only). The core claim of our study emphasizes how integrating both H&E and IHC whole-slide images (WSIs) outperforms single-stain models. While we acknowledge that benchmarking our H&E-only component on a publicly available dataset—such as TCGA CRC—would enable broader comparisons with the literature, it would reflect only a subset of our proposed framework and therefore not fully capture the advantages conferred by dual staining.

Furthermore, no publicly available dataset currently provides paired H&E and IHC WSIs required to validate the dual-modality aspect of our model. Datasets like TCGA do offer high-quality H&E WSIs but lack the corresponding IHC slides or clinical-grade annotations necessary to evaluate the synergy between the two modalities. Consequently, external validation on these datasets would not address the primary focus of our study: the benefit of integrating H&E and IHC data.

We fully recognize the value of external validation and plan to explore benchmarking our H&E-only component on public datasets in future work. However, for the present study, we focus on advancing the dual-modality approach and have acknowledged in the manuscript that the absence of publicly available dual-modality datasets is a limitation. Despite these constraints, DuoHistoNet demonstrated robust performance across our internal validation sets, particularly in MSI and MMRd prediction, highlighting its potential clinical utility. We will continue to seek additional avenues for external validation as more comprehensive multimodal datasets become available.

6. I am confused by the sensitivity / false negative rate calculation on pg 8, 1158 - 167.

How is it possible that the sensitivity increases from 98%  0.99% while false negative rate is reduced from 0.1% to 0.05%? What is the meaning of "This allowed for the accurate detection of 57% of Microsatellite Stable (MSS) patients", is this referring to the NPV?

Response: The key lies in recognizing that sensitivity focuses only on **positive** cases ($TP / [TP + FN]$), whereas a **false negative percentage** of 0.1% or 0.05% refers to $FN / total\ samples$. Even a small decrease in the absolute number of false negatives can cause a noticeable jump in sensitivity but only a modest change when spread across **all** samples.

In Figure 2d (MSI plot), we illustrate this trade-off. The x-axis shows the false negative percentage ($FN / total\ samples$), the left y-axis shows how many patients are correctly identified as MSS (true negatives / total samples), and the right y-axis indicates sensitivity ($TP / [TP + FN]$). Adjusting the classification threshold changes these metrics.

Calculation Example (Threshold = 0.04):

- Total Samples: 20,410
- True Negatives (TN): 11,698
- True Positives (TP): 1,355
- False Negatives (FN): 20
- False Positives (FP): 7,337

From these values:

- Sensitivity $\approx 1,355 \div (1,355 + 20) \approx 98.55\%$
- False Negative Percentage $\approx 20 \div 20,410 \approx 0.098\%$ (about 0.1%)

- Proportion of MSS Patients Correctly Identified $\approx 11,698 \div 20,410 \approx 57\%$

For the second question, we acknowledge that our initial wording may have caused confusion. To clarify, the 57% refers to the proportion of the total cohort (20,410 patients) that were correctly classified as MSS, rather than a performance metric such as sensitivity or negative predictive value (NPV). Specifically, among the 20,410 patients, 11,698 were correctly classified as MSS, accounting for 57% of the entire cohort. We have also provided Supplementary Data for further details.

To ensure clarity, we have adjusted our wording in the revised manuscript (lines 169–171, 174–175) to explicitly state that this percentage represents the proportion of the total cohort classified as MSS rather than a performance metric.

Reviewer #3 Comments and Responses

Reviewer #3 (Remarks to the Author):

Overall, this is a well-written manuscript and a nice study, which tackles a timely and important topic. The dataset is diverse and huge, the pipeline robust and well-designed. However, there are some questions/points that need to be addressed in a revised version.

Major Comments

1. *The authors should clarify already in the results section which IHC stains were used (MLH1/PMS2/MSH2/MSH6?).*

Response: We used MLH1 IHC staining for our study, we also address that in Results section.

2. *Innocently provocative point: If MMR stains are available, why should we “predict” MMR status? For a pathologist, it takes not much time to analyze such an IHC. Do the authors have any idea how their approach deals with difficult and/rare staining patterns such as described here DOI: 10.1038/s41379-022-01109-4, especially clonal loss/heterogenous staining (please also include this in the methods section how these rare staining patterns were classified, lines 365-369). Moreover, it would be highly interesting to look which cases where mis-classified by the model even though it had H&E as well as IHC data. As a pathologist, I would be quite interested to look into those cases more closely and think this is also of broader interest. Maybe the authors could make these cases publicly available with their publication and/or include representative cases in their supplement. A pathologic review of the misclassified cases should be performed.*

Response: We fully acknowledge that for an experienced pathologist, MMR IHC interpretation is typically straightforward. Nevertheless, scalability remains a challenge in high-volume pathology settings, particularly when resources or personnel are limited. Moreover, traditional manual assessment usually requires evaluating all four MMR proteins to accurately determine MMR status. By contrast, our approach can leverage a single MMR IHC stain in conjunction with H&E data, offering greater flexibility when one or more IHC stains are unavailable, fail to work properly, or are of suboptimal quality. Regarding difficult or rare staining patterns—such as clonal loss or heterogeneous staining—we agree that these present important and complex diagnostic challenges.

Our current dataset does not include detailed annotations on these specific rare patterns, limiting our ability to perform a robust analysis of how the model behaves in such scenarios. We appreciate the suggestion to incorporate more nuanced IHC phenotypes and plan to address this in future studies.

We also understand the interest in examining misclassified cases in greater detail. In our internal analyses, misclassifications were often attributable to:

- **Poor Sample Quality:** Technical issues such as suboptimal deparaffinization can compromise staining.
- **Limited Tissue Area or Low Tumor Percentage:** Insufficient tumor coverage may hinder reliable feature extraction.
- **CRC-MMRd/CRC-MSI Cases Retaining MLH1:** Our model uses only the MLH1 IHC slide; thus, samples retaining MLH1 but losing a different MMR protein were at risk of misclassification.

For misclassified cases that do not fall into the previously mentioned categories, we plan to conduct more detailed pathologic reviews as resources permit. We recognize that a thorough analysis of these cases would be of broad interest, and we have acknowledged our current limitation of not performing a comprehensive pathologic review of the heatmaps in the updated Discussion section. To partially address this gap, we have included representative heatmaps of misclassified cases in **Supplementary Figure 5-7**. Additionally, as stated in the “Method/Data Availability” section, we are open to facilitating access to our data for independent evaluation, subject to patient privacy constraints. We hope these measures will help clarify the specific reasons for misclassifications and underscore the potential for further refinement of our method.

3. What did the authors see in their heatmaps (lines 169-172)? Was a pathological review performed? Any new insights? As they describe that this has the potential to identify new patterns? In Supplementary Figure 1a the H&E shows an adenocarcinoma with gland-forming morphology (NOS) and retained MMR staining (which MMR marker IHC is displayed here?). In Supplementary Figure 1b a potentially medullary CRC is displayed which is a typical MSI-like morphology. Moreover, the normal mucosa shows retained MMR expression and is not highlighted in the heatmap. Comparing MSI prediction heatmaps and IHC stains, especially in heterogenous cases, could be highly interesting and could prove that DL-based MSI predictors can resolve intratumoral heterogeneity on a biologic basis. For references on these topics: doi: 10.1038/modpathol.2016.198, doi: 10.1136/gutjnl-2019-319866)

Response: Thank you for the thoughtful questions and for providing the relevant references. We share your enthusiasm for exploring how DL-based models can illuminate intratumoral heterogeneity and reveal novel histopathologic patterns.

In our study, we performed a preliminary pathologic review of the generated heatmaps, which highlighted specific tumor regions on both H&E and IHC slides. More specifically, three types of heatmaps were generated for pathologic review:

- **Attention Heatmap (ATTN):** Highlights the regions or features the model focuses on when predicting the biomarker status. Bright areas indicate where the model assigns higher importance.

- **Classification Heatmap (CLS):** Shows the predicted probability of biomarker presence for different regions. High-intensity areas indicate a strong association with biomarker positivity.
- **Contribution Heatmap (ATTNxCLS):** Combines attention and classification heatmaps to reveal which regions most influence the biomarker prediction. Bright areas indicate high score of combined attention and classification, suggesting they are key for biomarker status determination.

Below is a brief summary of our observations based on the contribution heatmap heatmaps:

- **H&E Heatmaps**

CRC Cases (MMRp/MMRd): Highlighted tiles often showed areas of moderate to poor differentiation, pleomorphic nuclei, and brisk mitotic or apoptotic activity. In MMRd (MSI) cases, the model frequently emphasized tiles at the tumor-stromal interface with dense inflammatory infiltrates.

PD-L1 Cases: In PD-L1-positive tumors, the model often focused on regions exhibiting nuclear pleomorphism and poor differentiation. By contrast, in PD-L1-negative tumors, highlighted regions were commonly found along the leading edge, possibly reflecting subtle morphologic cues indicative of tumor-host interaction.

- **IHC Heatmaps**

CRC Cases (MMRp/MMRd): Highlighted tiles included both regions with significant IHC staining and areas with absent or reduced staining. In MSI cases, tiles with loss of MMR protein expression were emphasized, consistent with the model's focus on abnormal immunostaining.

PD-L1 Cases: The model highlighted areas of strong PD-L1 staining but occasionally flagged regions with minimal or no staining. This suggests that additional morphologic features—beyond the mere presence or intensity of staining—may influence how the network assigns attention.

With respect to **Figure 1b**, the model did not highlight certain areas despite retained MMR staining, indicating that it prioritized tumor-associated features rather than relying solely on staining intensity or presence.

Overall, we did not identify any completely novel or previously undescribed morphologic patterns in these heatmaps. However, we believe this approach holds promise for detecting intratumoral heterogeneity—particularly in cases where mixed or “patchy” immunostaining might be present. We plan to investigate this further in future work by expanding our analysis to include more heterogeneous tumor samples and correlating heatmap findings with detailed pathologic assessments, as you suggested.

Finally, while our initial pathologic review provides insights into the model's mechanisms, a more extensive and systematic evaluation would be valuable. We are continuing to refine our methods to capture and interpret fine-grained features (e.g., nuclear measurements, inflammatory cell density, architectural patterns), which may

help us better understand how DL-based MSI prediction tools can identify and resolve intratumoral heterogeneity on a biologic basis.

We appreciate the reviewer's interest and suggestions and look forward to exploring these directions in greater detail in future analyses.

4. *Lines 127-128: MLH1 promotor hypermethylation and consecutive loss of MLH1 expression is the most common reason for dMMR (doi: 10.1053/j.gastro.2009.12.064) but I think the phrase “due to its high concordance with MMRd status” is a bit misleading. If a case shows MLH1 expression loss, it is per se dMMR. As MMR proteins function as heterodimers (MLH1/PMS2; MSH2/MSH6), other staining patterns can be observed – even the loss in both heterodimers (<https://doi.org/10.3389/fonc.2022.1019798>).*

Response: We agree with reviewer's comments and have rephrase our wording in line 133-134 accordingly. We also recognize that MMR proteins function as heterodimers (MLH1/PMS2 and MSH2/MSH6), and that different patterns of staining loss—including combined losses—may occur.

3. *How do the authors explain the poorer performance of MMR prediction when focusing on metastatic specimens (resection specimens and/or biopsies)? In the literature, it is known that performance drops when trying to predict MSI status from biopsy material and/or metastases. However, when integrating IHC this should be improved. Do the authors have further ideas how MSI prediction on biopsies and/or metastatic sites could be improved in the future, as these two indications are very*

important: neoadjuvant IO-therapy (NICHE2, DOI: 10.1056/NEJMoa2400634) and IO-therapy in the palliative/metastatic setting.

Response: Thank you for highlighting the importance of predicting MSI/MMR status in metastatic specimens and biopsies, particularly in the context of neoadjuvant and palliative immunotherapy. In our study, we observed that the performance of the H&E-only model dropped more substantially for metastatic samples (AUROC 0.940 vs. 0.852 for MMRd prediction and 0.948 vs. 0.899 for MSI prediction in primary versus metastatic specimens). By contrast, the IHC-only model showed less decline (AUROC 0.954 vs. 0.917 for MMRd and 0.954 vs. 0.936 for MSI, respectively).

Nonetheless, our current analysis shows that the duet model—which integrates both H&E and IHC images—performs better than either the H&E-only or IHC-only models. This suggests that incorporating IHC data can partially mitigate the challenges of predicting MSI/MMR status in metastatic specimens.

There are several possible reasons for the decreased accuracy observed in metastatic samples, particularly for the H&E-only model:

- a. **Greater Histologic Heterogeneity:** Metastatic lesions often exhibit more variable morphology, making it harder for a morphology-focused model to reliably recognize patterns indicative of dMMR or MSI.
- b. **Sampling Differences:** Biopsies or partial resection specimens may contain less tumor tissue and a higher proportion of non-tumor elements, thereby reducing the amount of informative tumor regions.

Looking ahead, we see opportunities to improve MSI/MMR prediction in biopsies and metastatic specimens.

- a. Diverse datasets: Incorporating biopsies and metastatic specimens from a broader spectrum of malignancies may increase the robustness and generalizability of the model.
- b. Enhanced Multimodal Inputs:
 - Clinical/Pathologic Context: Incorporating text-based metadata—such as the anatomic site of metastasis, resection vs. biopsy status, or relevant pathology findings—could help the model adapt to the nuances of metastatic lesions and small biopsy specimens.
 - Molecular Profiles: Integrating additional molecular data (e.g., targeted gene panels or RNA sequencing) may help the algorithm pick up subtle signals not fully captured by histopathology alone.
- c. Advanced Image Processing:
 - Multiscale feature extraction: our current model extracts features from 10x WSIs, and multiscale feature extraction might capture more nuanced spatial information.
 - Augmented Training: Generating “biopsy-like” patches from larger resection specimens could mimic the appearance and tissue constraints of true biopsies, helping the model learn to cope with smaller tissue areas.
- d. Foundation Models: Leveraging recently developed large-scale “foundation models” can help the network extract robust, generalizable features, which might be particularly beneficial for understudied or heterogeneous metastases.

5. *The authors argue that their model can reduce the necessity for NGS-based MSI testing. This is correct. However, this is also correct for a solely HE-based MSI deep-learning predictor such as MSIntuit by Owkin (in this scenario you even save money and time for IHC). Moreover, by performing (large panel) NGS testing that allows MSI assessment you gain further important information on driver mutations such as KRAS, BRAF etc. but also on rare but important mutations in CRC such as POLE. Could the authors please comment on this or even include their thoughts into the manuscript. Could the authors also bring up arguments why their pipeline is superior to e.g. MSIntuit – if they believe this is the case.*

Response: We appreciate the reviewer’s insightful comments. We acknowledge that NGS-based testing provides extensive information—not only on MSI status but also on critical driver mutations (e.g., KRAS, BRAF) and rare yet clinically significant alterations such as POLE mutations. However, in resource-limited settings, particularly in less developed countries, an H&E-based deep-learning predictor for MSI (or our proposed pipeline) offers a more accessible and cost-effective option compared to NGS or even a solely H&E-based model such as MSIntuit.

To directly compare performance, MSIntuit reports a sensitivity of 0.96–0.98 with a specificity of 0.46–0.471 (Saillard, Dubois et al. 2023). In comparison, our H&E WSI-based model achieved a specificity of 42.8% at a sensitivity of 98% and a specificity of 62.4% at a sensitivity of 96% (Supplementary Data 4). Our DuoHistoNet model further improved performance, achieving a specificity of 65.1%

at a sensitivity of 98% and a specificity of 78.3% at a sensitivity of 96% (Supplementary Data 2). This indicates that our dual-modality approach provides pathology laboratories with a more effective tool for MSI pre-screening. It achieves higher specificity than MSIntuit while maintaining high sensitivity.

While a model like MSIntuit might reduce the need for IHC and thereby save on costs and time, our pipeline provides complementary diagnostic information by leveraging both H&E and IHC data. This multimodal integration not only enhances diagnostic performance but also serves as an effective pre-screening tool. By accurately identifying patients with MSI/MMRd, our approach can help prioritize those who truly require further molecular analysis through NGS testing. This strategy ultimately optimizes resource allocation and reduces the overall need for expensive and time-consuming NGS tests, without compromising diagnostic accuracy.

In summary, our dual-modality pipeline is advantageous over a solely H&E-based predictor like MSIntuit, as it significantly improves specificity while maintaining high sensitivity. Moreover, it offers a cost-effective pre-screening method that complements the comprehensive information gained from NGS testing.

References:

Saillard C, et al. Validation of MSIntuit as an AI-based pre-screening tool for MSI detection from colorectal cancer histology slides. Nature Communications 14, 6695 (2023).

6. It is great that the authors include survival analysis. However, if survival data is available, the authors could try to predict survival and/or treatment response directly from HE and IHC, and without the necessity/detour of predicting MSI/dMMR status.

Response: We appreciate the reviewer's suggestion regarding the direct prediction of survival and/or treatment response from H&E and IHC images, bypassing MSI/dMMR status as an intermediate step. In our ongoing studies involving non-small cell lung cancer and breast cancer patients, we are actively exploring deep-learning models that directly predict clinical outcomes and treatment response. While these efforts are still in progress, initial results are promising, and we plan to present more comprehensive findings on direct prediction models in our future publications.

7. The HE-model in breast cancer surpassed the actual pathologist-scored CPS in significance. What are the potential reasons for this? This is interesting. To the authors think this has a biologic background. Were heatmaps revisited? Were the actual pathologist scores re-viewed if they were correct?

Response: We appreciate the reviewer's interest in why the H&E-based model for breast cancer surpassed the pathologist-scored CPS in terms of survival prediction. In our current analysis, patients were stratified as PD-L1–positive or –negative using a cutoff of CPS ≥ 10 . While this threshold has regulatory and clinical precedent in triple-negative breast cancer (TNBC) for immunotherapy eligibility, it might not be the optimal cutoff for predicting overall survival (OS) or time on treatment (TOT) in our specific cohort. This hypothesis would require further validation, potentially by testing

alternative CPS thresholds in a larger, independent dataset. As part of our internal quality control, a pathologist revisited a subset of PD-L1 cases and confirmed the original CPS assessments, suggesting that the scoring itself was consistent.

Nonetheless, intratumoral heterogeneity, variations in pre-analytical conditions, and differences in patient populations may all contribute to discrepancies between the deep-learning model's performance and the pathologist-assigned CPS. Furthermore, the deep-learning model may be capturing additional morphologic or microenvironmental features not fully accounted for by the CPS alone.

Minor comments

1. *Line 51-52: MSI status assessment most often relies on PCR-based methods, but NGS is becoming more prevalent. For MMR testing, IHC is of course correct.*

Response: We appreciate the reviewer's clarification. We have revised lines 50–53 to state that MSI status is most often assessed via PCR-based methods, although NGS is increasingly utilized for comprehensive molecular profiling.

2. *Lines 61-67: For an overview of DL-based techniques in CRC refer to the extensive literature on this, e.g. doi: 10.1159/000539678.*

Response: We appreciate the reviewer's recommendation. We have now cited the suggested reference (doi: 10.1159/000539678) in the updated main text on line 62. We note that this article was published after our initial submission and thus was not included in the original manuscript.

3. *Patient Characteristics: I assume that the CRC MMRd cohort was tested via IHC and the CRC MSI cohort via PCR? But could you please clarify. If possible, please provide more detailed clinicopathological characteristics of the cohorts (pTNM stage, L/V/Pn, Budding, hormone receptors/HER2/Ki67 status, if available). CRC as well as BRCA are really heterogenous diseases.*

Response: We appreciate the reviewer's request for more detailed clinicopathological information. In our study, the CRC MMRd cohort was tested via IHC, while the CRC MSI cohort underwent NGS analysis. We fully acknowledge that both CRC and breast cancer are highly heterogeneous diseases; however, additional clinicopathological characteristics were not available for our current cohorts. We plan to collect and include more comprehensive clinical and pathological data in future studies, which will allow for deeper subgroup analyses and a more nuanced understanding of tumor heterogeneity.

4. *Why is the model framework "revised"? Is there any prior version? If so, please cite/refer to this.*

Response: We apologize for any confusion caused by the term "revised." There is no prior version of our model framework, and we have removed that wording from the manuscript to avoid misunderstanding.

5. *Please tone down in lines 108-110: Other studies also use ViT for HE and IHC analysis (just as example <https://doi.org/10.1016/j.media.2024.103289>). Of course, the field is here rapidly evolving.*

Response: Thank you for providing the reference and for your feedback. In the updated main text (lines 109–112), we have removed the claim of being the inaugural application and clarified the unique aspects of our approach, reflecting the rapidly evolving nature of this field.

6. Lines 121-123: Different algorithms have shown to predict MSI, however here only the Wagner et al. paper on transformer-based MSI prediction is cited (in Wagner et al. AttMIL, CNN etc are also included). However, when taking about different multiple studies the authors should provide adequate references.

Response: We appreciate the reviewer’s feedback. We have now included additional references (lines 125–130) that highlight multiple studies employing various deep-learning architectures for MSI prediction. These updates supplement the previously cited work by Wagner et al. (Wagner, Reisenbuchler et al. 2023), ensuring a more comprehensive overview of the current literature.

References

Wagner, S.J., et al. Transformer-based biomarker prediction from colorectal cancer histology: A large-scale multicentric study. Cancer Cell 41, 1650-1661 e1654 (2023).

7. Lines 124-125: Is mentioning a CT-based study here necessary? If so, the unique approach with predicting MSI status not from tissue biopsies but non-invasively via imaging data should be made clear.

Response: We appreciate the reviewer's observation. As our primary focus is on histopathology-based analyses, referencing a CT-based study is not essential. Therefore, we have removed that sentence in the updated main text to maintain clarity on our main objectives.

8. *The risk tables in the Kaplan Meier curves are quite small.*

Response: We have increased the font size of the risk tables to the extent possible, given the layout constraints. We hope this improvement enhances readability while maintaining the overall formatting of the figures.

Please note that additional edits were made to meet editorial requirements. All changes are highlighted in the tracked version of the manuscript.

Reviewer #1

Comment:

No comment.

Response:

We thank the reviewer for taking the time to evaluate our manuscript.

Reviewer #2

Comment:

Thank you for updating the manuscript with additional experiments and clarification – they have addressed all my concerns.

Response:

We thank the reviewer for their careful review and positive feedback. We are glad that the revisions have satisfactorily addressed all concerns.

Reviewer #3

Comment:

As the authors only used MLH1-staining for MMR-status, this should be stated as major limitation. They also state that this may lead to misclassification (retained MLH1-expression leading to pMMR prediction etc) despite of immunohistochemical loss of PMS2, or MSH2 or MSH6. Even in two stain approaches, pathologists use PMS2/MSH6 staining (<https://pubmed.ncbi.nlm.nih.gov/29967423/>). The authors should do follow-up studies with all immunohistochemical markers.

Response:

We thank the reviewer for this insightful and constructive comment. In response, we have explicitly acknowledged the limitation of assessing MMR status using only MLH1 IHC. As

noted, relying solely on MLH1 may result in misclassification when MLH1 is retained but other MMR proteins such as PMS2, MSH2, or MSH6 are lost. We now cite Pearlman et al. (Mod Pathol, 2018), which highlights the risk of underdetection using two-stain approaches.

To address this limitation, we have revised the Discussion section to emphasize the need for comprehensive IHC evaluation. We also clarify that future studies will incorporate all four recommended IHC markers—MLH1, PMS2, MSH2, and MSH6—into model training to improve diagnostic precision and clinical applicability.

Revision Location:

Please see the revised Discussion section at line 593-609.